# FedDebias: Reducing the Local Learning Bias Improves Federated Learning on Heterogeneous Data

## Abstract

Federated Learning (FL) is a machine learning paradigm that learns from data kept locally to safeguard the privacy of clients, whereas local SGD is typically employed on the clients' devices to improve communication efficiency. However, such a scheme is currently constrained by the slow and unstable convergence induced by clients' heterogeneous data. In this work, we identify three under-explored phenomena of the biased local learning that may explain these challenges caused by local updates in supervised FL. As a remedy, we propose FedDebias, a novel unified algorithm that reduces the local learning bias on features and classifiers to tackle these challenges. FedDebias consists of two components: The first component alleviates the bias in the local classifiers by balancing the output distribution of models. The second component learns client invariant features that are close to global features but considerably distinct from those learned from other input distributions. In a series of experiments, we show that FedDebias consistently outperforms other SOTA FL and domain generalization (DG) baselines, in which both two components have individual performance gains.

## 1 Introduction

Federated Learning (FL) is an emerging privacy-preserving distributed machine learning paradigm. The model is transmitted to the clients by the server, and when the clients have completed local training, the parameter updates are sent back to the server for integration. Clients are not required to provide local raw data during this procedure, maintaining their privacy. As the workhorse algorithm in FL, FedAvg (McMahan et al., 2016) proposes local SGD to improve communication efficiency. However, the considerable heterogeneity between local client datasets leads to inconsistent local updates and hinders convergence.

Several studies propose variance reduction methods (Karimireddy et al., 2019; Das et al., 2020), or suggest regularizing local updates towards global models (Li et al., 2018b; 2021) to tackle this issue. Almost all these existing works directly regularize models by utilizing the global model collected from previous rounds to reduce the variance or minimize the distance between global and local models (Li et al., 2018b; 2021). However, it is hard to balance the trade-offs between optimization and regularization to perform well, and data heterogeneity remains an open question in the community, as justified by the limited performance gain, e.g. in our Table 1.

To this end, we begin by revisiting and reinterpreting the issues caused by data heterogeneity and local updates. We identify three pitfalls of FL, termed *local learning bias*, from the perspective of representation learning[1]: 1) Biased local classifiers are unable to effectively classify unseen data (in Figure 1(a)), due to the shifted decision boundaries dominated by local class distributions; 2) Local features (extracted by a local model) differ significantly from global features (similarly extracted by a centralized global model), even for the same input data. (c.f. Figure 1(b)); and 3) Local features, even for data from different classes, are close to each other and cannot be accurately distinguished (c.f. Figure 1(b)).

As a remedy, we propose FedDebias, a unified method that leverages a globally shared pseudo-data and two key algorithmic components to simultaneously address the three difficulties outlined above. The first component of FedDebias alleviates the first difficulty by forcing the output distribution of

---

[1]Please refer to section 3 for more justification about the existence of our observations.

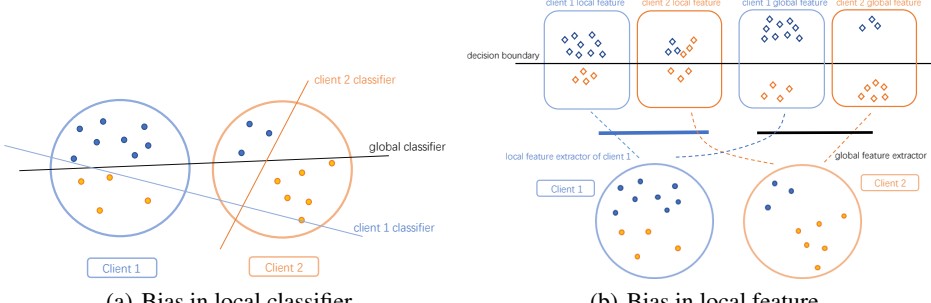

(a) Bias in local classifier.  (b) Bias in local feature.

Figure 1: **Observation for learning bias**: three pitfalls of FL on heterogeneous data with local updates. There are two clients in the figure (denoted by two colors), and each has two classes of data (red and blue points). **Figure 1(a)**: Client 1's decision boundary cannot accurately classify data samples from client 2. **Figure 1(b)**: The difference between features extracted by client 1's local feature extractor and global feature extractor is sustainable large. However, client 2's local feature is close enough to client 1's, even for input data from different data distributions/clients.

the pseudo-data to be close to the global prior distribution. The second component of FedDebias is designed for the second and third difficulties. In order to tackle the last two difficulties simultaneously, we develop a min-max contrastive learning method to learn client invariant local features. More precisely, instead of directly minimizing the distance between global and local features, we design a two-stage algorithm. The first stage learns a projection space—an operation that can maximize the difference between global and local features but minimize local features of different inputs—to distinguish the features of two types. The second stage then debiases the features by leveraging the trained projection space to enforce learned features that are farther from local features and closer to global features.

We examine the performance of FedDebias and compare it with other FL and domain generalization baselines on RotatedMNIST, CIFAR10, and CIFAR100. Numerical results show that FedDebias consistently outperforms other algorithms by a large margin on mean accuracy and convergence speed. Furthermore, both components have individual performance gains, and the combined approach FedDebias yields the best results.

**Contributions**

- We propose FedDebias, a unified algorithm that leverages pseudo-data to reduce the learning bias on local features and classifiers. We design two orthogonal key components of FedDebias to complement each other to improve the learning quality of clients with heterogeneous data.
- FedDebias considerably outperforms other FL and domain generalization (DG) baselines, as justified by extensive numerical evaluation.

## 2 RELATED WORKS

**Federated Learning (FL).**  As the de facto FL algorithm, McMahan et al. (2016); Lin et al. (2020b) propose to use local SGD steps to alleviate the communication bottleneck. However, the objective inconsistency caused by the local data heterogeneity considerably hinders the convergence of FL algorithms (Li et al., 2018b; Wang et al., 2020; Karimireddy et al., 2019; 2020; Guo et al., 2021). To address the issue of heterogeneity in FL, a series of projects has been proposed. FedProx (Li et al., 2018b) incorporates a proximal term into local objective functions to reduce the gap between the local and global models. SCAFFOLD (Karimireddy et al., 2019) adopts the variance reduction method on local updates, and Mime (Karimireddy et al., 2020) increases convergence speed by adding global momentum to global updates.

**Data Augmentation in FL.**  To reduce data heterogeneity, some data-based approaches suggest sharing a global dataset among clients and combining global datasets with local datasets (Tuor et al., 2021; Yoshida et al., 2019). Some knowledge distillation-based methods also require a global dataset (Lin et al., 2020a; Li & Wang, 2019), which is used to transfer knowledge from local models (teachers) to global models (students). Considering the impractical of sharing the global datasets in FL settings, some recent research use proxy datasets with augmentation techniques. Astraea (Duan et al., 2019) uses local augmentation to create a globally balanced distribution. XorMixFL (Shin et al., 2020) encodes a couple of local data and decodes it on the server using the XOR operator. FedMix (Yoon et al., 2021b) creates the privacy-protected augmentation data by averaging local

batches and then applying Mixup in local iterations. VHL (Tang et al., 2022) relies on the created virtual data with labels and forces the local features to be close to the features of same-class virtual data. Our framework significantly outperforms VHL; unlike VHL, our solution has no label constraint and uses much less pseudo-data than VHL.

**Distribution Robust FL.** Domain generalization is a well-studied field, aiming to learn domain-robust models that perform well on unknown distributions. Some methods apply domain robust optimization methods (Sagawa et al., 2019; Hu & Hong, 2013; Michel et al., 2021) to minimize the worst-case empirical error, and others propose to learn domain invariant features (Ganin et al., 2015; Li et al., 2018c;a; Sun & Saenko, 2016) by minimizing the distance of features from different domains. By treating each client as a domain, some existing works tackle the FL problem as a domain generalization problem. Several methods include optimizing the weights of different clients to lower the worst empirical error among all clients (Mohri et al., 2019; Deng et al., 2021). Huang et al. (2021) assumes each client has two local datasets with a different distribution, and the robustness is obtained by balancing the two local datasets. Xie et al. (2021) proposes collecting gradients from one segment of clients first, then combining them as a global gradient to reduce variance in the other segments. Reisizadeh et al. (2020) assumes the local distribution is perturbed by an affine function, i.e., from $x$ to $Ax + b$. There are also some methods that aim to learn client invariant features (Peng et al., 2019; Wang et al., 2022; Shen et al., 2021; Sun et al., 2022; Gan et al., 2021). However, these methods are designed to learn a model that can perform well on unseen deployment distributions that differ from the (seen) clients' local distributions, which is beyond the scope of this paper.

Recently, Moon (Li et al., 2021) has proposed to employ contrastive loss to reduce the distance between global and local features. However, their projection layer is only used as part of the feature extractor, and cannot contribute to distinguish the local and global features—a crucial step identified by our investigation for a better model performance.

## 3 THE PITFALLS OF FL ON HETEROGENEOUS DATA DISTRIBUTIONS

**FL and local SGD.** FL is an emerging learning paradigm that supposes learning on various clients while clients can not exchange data to protect users' privacy. Learning occurs locally on the clients, while the server collects and aggregates gradient updates from the clients. The standard FL considers the following problem:

$$f^* = \min_{\boldsymbol{\omega} \in R^d} \left[ f(\boldsymbol{\omega}) = \sum_{i=1}^{N} p_i f_i(\boldsymbol{\omega}) \right], \tag{1}$$

where $f_i(\boldsymbol{\omega})$ is the local objective function of client $i$, and $p_i$ is the weight for $f_i(\boldsymbol{\omega})$. In practice, we set $p_i = |D_i|/|D|$ by default, where $D_i$ is the local dataset of client $i$ and $D$ is the combination of all local datasets. The global objective function $f(\boldsymbol{\omega})$ aims to find $\boldsymbol{\omega}$ that can perform well on all clients. In the training process of FL, the communication cost between client and server has become an essential factor affecting the training efficiency. Therefore, local SGD (McMahan et al., 2016) has been proposed to reduce the communication round. In local SGD, clients perform multiple local steps before synchronizing to the server in each communication round.

**The negative impact of local update steps.** Despite the success of local SGD, the non-iid nature of clients' local data leads to local gradient inconsistency, which will slow down the convergence (Li et al., 2018b; Karimireddy et al., 2019). A series of studies have proposed several methods for client heterogeneity to address this issue. One natural idea considers using the global gradient/model of previous rounds during the local updates to reduce variance or minimize the distance between the global and local model (Karimireddy et al., 2019; 2020; Li et al., 2018b; 2021). However, the performance of such algorithms is limited in our challenging scenarios (as we shown in Table 1). Using FedProx (Li et al., 2018b) as an example, setting larger weights for proximal terms will hinder the further optimization steps of the local model, while setting a small weight will result in a marginal improvement of FedProx over FedAvg.

**Bias caused by local updates.** To mitigate the negative impact of local updates, we first identify the pitfalls of FL on heterogeneous data with a sufficient number of local updates and then design the algorithms to address the issues caused by the local updates.

The pitfalls can be justified by a toy experiment. More precisely, we divide the MNIST dataset into two sets. The first dataset, denoted by $X_1$, contains the 5 classes 0-4. The other dataset, denoted by $X_2$, contains the remaining five classes. Then we train a CNN model on $X_1$ for 10 epochs and store

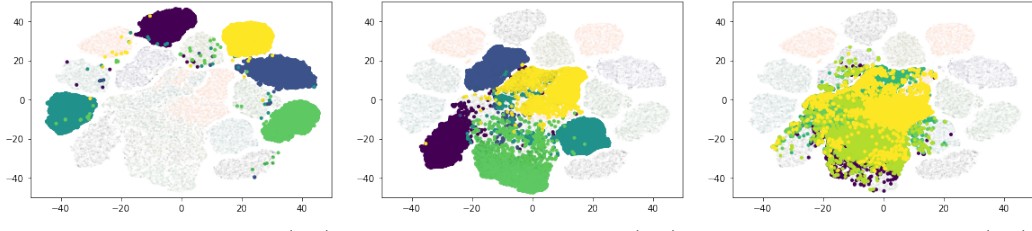

(a) Global feature of $X_1$, $F_g(X_1)$    (b) Local feature of $X_1$, $F_1(X_1)$    (c) Local feature of $X_2$, $F_1(X_2)$

Figure 2: **Observation for biased local features** on a shared t-SNE projection space. *Local updates will cause:* ● *Large difference in local and global features for the same input data.* Colored points in sub-figures (a) & (b) denote the global and local features of data from $X_1$, and the same color indicates data from the same class. Notice that even for data from the same class (same color), the global and local features are clustered into two distinct groups, implying a considerable distance between global and local features even for the same input data distribution. ● *High similarity of local features for different inputs.* Notice from sub-figure (b) & (c) that $X_1$ and $X_2$ are two disjoint datasets (no data from the same class). However, the local features of $X_1$ and $X_2$ are clustered into the same group by t-SNE, indicating the relatively small distance between local features of different classes.

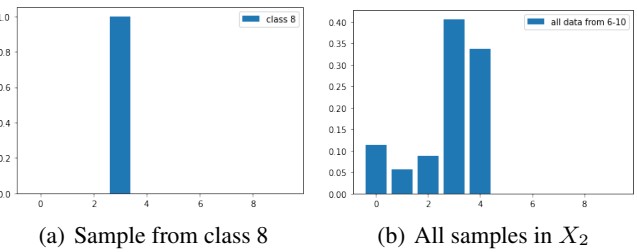

(a) Sample from class 8        (b) All samples in $X_2$

Figure 3: **Observation for biased local classifiers**: *the output distribution of the local classifiers will be dominated by the local class distribution*. The model is trained on data $X_1$ and tested on data $X_2$. The sub-figure (a) illustrates the model output distribution of a sample belonging to Class 8. The sub-figure (b) shows the total prediction distribution of all samples in $X_2$. Results show that *the biased local model will classify all samples into classes that are only present in the $X_1$*.

the feature extractor of the local model as $F_1$. We also train a global model on the mixture of $X_1$ and $X_2$ (equal to centralized training) and store the feature extractor as $F_g$. We use $F_g$ as the ground truth model because it works well over the entire dataset and does not suffer from local updates.

**Example 3.1** (Observation for biased local features). *Figures 2(a) and 2(b) show that* local features differ from global features for the same input, *and Figures 2(b) and 2(c) show that* local features are similar even for different input distributions. *We define this observation as the "biased local feature". In detail, we calculate $F_1(X_1)$, $F_1(X_2)$, $F_g(X_1)$, and $F_g(X_2)$, and use t-SNE to project all the features to the same 2D space.* [2] *We can observe that the local features of data in $X_2$ are so close to local features of data in $X_1$, and it is non-trivial to tell which category the current input belongs to by merely looking at the local features.*

**Example 3.2** (Observation for biased local classifiers). *Figure 3 shows the output of local model on data $X_2$, where all data in $X_2$ are incorrectly categorized into classes 0 to 4 of $X_1$. The observation, i.e., data from classes that are absent from local datasets cannot be correctly classified by the local classifiers, refers to the "biased local classifiers". More precisely, Figure 3(a) shows the prediction result of one sample (class 8) and Figure 3(b) shows the predicted distribution of all samples in $X_2$.*

Based on Example 3.1 and Example 3.2, we summarize our observations and introduce the formal definition of "local learning bias" caused by local updates:

**Definition 3.3** (Local Learning Bias). *We define the local learning bias below:*

● Biased local feature: *For local feature extractor $F_i(\cdot)$, and centralized trained global feature extractor $F_g(\cdot)$, we have: 1) Given the data input $X$, $F_i(X)$ could deviate largely from $F_g(X)$. 2) Given the input from different data distributions $X_1$ and $X_2$, $F_i(X_1)$ could be very similar or almost identical to $F_i(X_2)$.*

● Biased local classifier: *After a sufficient number of iterations, local models classify all samples into only the classes that appeared in the local datasets.*

---

[2] We provide the results after using FedDebias in Appendix C.3.

It is worth to note that some related works also discussed the learning bias(Karimireddy et al., 2019; Li et al., 2018b; 2021). However, we have an inherent difference compared with previous works. 1) FedProx defines the local drifts as the difference between model weights $\|\omega_g - \omega_i\|$, and SCAFFOLD considers gradient difference as client drifts. Despite the theoretical success of these methods, these two methods usually have minor improvements on deep models (Tang et al., 2022; Li et al., 2021; Yoon et al., 2021a; Chen & Chao, 2021; Luo et al., 2021). 2) Though MOON is a crucial first step that minimizes the distance between global and local features, its performance gain is still limited due to the improper methodology design (cf. Table 1).

## 4 FedDebias: Reducing Learning Bias in FL by Pseudo-data

Addressing the local learning bias is crucial to improving FL on heterogeneous data, due to the *bias* discussed in Definition 3.3. To this end, we propose FedDebias as shown in Figure 4, a novel framework that leverages the globally shared pseudo-data with two key components to reduce the local training bias, namely 1) reducing the local classifier's bias by balancing the output distribution of classifiers (component 1), and 2) an adversary contrastive scheme to learn unbiased local features (component 2).

### 4.1 Overview of the FedDebias

The learning procedure of FedDebias on each client $i$ involves the construction of a global pseudo-data (c.f. Section 4.2), followed by applying two key debias steps in a ***min-max*** approach to jointly form two components (c.f. Section 4.3 and 4.4) to reduce the bias in the classifier and feature, respectively. The min-max procedure of FedDebias can be interpreted as first projecting features onto spaces that can distinguish global and local feature best, then 1) minimizing the distance between the global and local features of pseudo-data and maximizing distance between local features of pseudo-data and local data; 2) minimize classification loss of both local data and pseudo-data:

**Max Step:**

$$\max_{\boldsymbol{\theta}} \mathcal{L}_{adv}(D_p, D_i) = \mathbb{E}_{\mathbf{x}_p \sim D_p, \mathbf{x} \sim D_i} \left[ \mathcal{L}_{con}(\mathbf{x}_p, \mathbf{x}, \boldsymbol{\phi}_g, \boldsymbol{\phi}_i, \boldsymbol{\theta}) \right] . \tag{2}$$

**Min Step:**

$$\min_{\boldsymbol{\phi}_i, \boldsymbol{\omega}} \mathcal{L}_{gen}(D_p, D_i) = \mathbb{E}_{(\mathbf{x}, \mathbf{y}) \sim D_i} \left[ \mathcal{L}_{cls}(\mathbf{x}, \mathbf{y}, \boldsymbol{\phi}_i, \boldsymbol{\omega}) \right] + \lambda \mathbb{E}_{\mathbf{x}_p \sim D_p} \left[ \mathcal{L}_{cls}(\mathbf{x}_p, \tilde{\mathbf{y}}_p, \boldsymbol{\phi}_i, \boldsymbol{\omega}) \right]$$

$$+ \mu \mathbb{E}_{\mathbf{x}_p \sim D_p, \mathbf{x} \sim D_i} \left[ \mathcal{L}_{con}(\mathbf{x}_p, \mathbf{x}, \boldsymbol{\phi}_g, \boldsymbol{\phi}_i, \boldsymbol{\theta}) \right] . \tag{3}$$

$\mathcal{L}_{cls}$ and $\mathcal{L}_{con}$ represent the cross-entropy loss and a contrastive loss (will be detailed in Section 4.4), respectively. $D_i$ denotes the distribution of local dataset at client $i$. $D_p$ is that of shared pseudo-dataset, where $\tilde{\mathbf{y}}_p$ is the pseudo-label of pseudo-data. The model is composed of a feature extractor $\boldsymbol{\phi}$ and a classifier $\boldsymbol{\omega}$, where the omitted subscript $i$ and $g$ correspond to the local client $i$ and global parameters, respectively (e.g., $\boldsymbol{\phi}_g$ denotes the feature extractors received from the server at the beginning of each communication round). We additionally use a projection layer $\boldsymbol{\theta}$ for the max step to project features onto spaces where global and local features have the largest dissimilarity.

Apart from the standard classification loss of local data in Equation (3), the second term aims to overcome the biased local classifier while the local feature is debiased by the third term.

The proposed FedDebias is summarized in Algorithm 1. The global communication part is the same as FedAvg, and the choice of synchronizing the new pseudo-data to clients in each round is optional[3].

### 4.2 Construction of the Pseudo-Data

The choice of the pseudo-data in our FedDebias framework is arbitrary. For ease of presentation and taking the communication cost into account, we showcase two construction approaches below and detail their performance gain over all other existing baselines in Section 5:

- **Random Sample Mean (RSM)**. Similar to the treatment in FedMix (Yoon et al., 2021b), one RSM sample of the pseudo-data is estimated through a weighted combination of a random subset of local samples, and the pseudo-label is set[4] to $\tilde{\mathbf{y}}_p = \frac{1}{C} \cdot \mathbf{1}$. Details can be found in Algorithm 2 of Appendix B.

---

[3] As shown in Figure 5(b), the communication-efficient variant of FedDebias—i.e., only transferring pseudo-data at the beginning of the FL training—is on par with the choice of frequent pseudo-data synchronization.

[4] We assume an uniform distribution for label and pseudo-data does not belong to any particular classes.

**Require:** Local datasets $D_1, \ldots, D_N$, pseudo dataset $D_p$ where $|D_p| = B$, and $B$ is the batch size, number of local iterations $K$, number of communication rounds $T$, number of clients chosen in each round $M$, weights used in designed loss $\lambda, \mu$, local learning rate $\eta$.

**Ensure:** Trained model $\boldsymbol{\omega}_T, \boldsymbol{\theta}_T, \boldsymbol{\phi}_T$.
1: Initialize $\boldsymbol{\omega}_0, \boldsymbol{\theta}_0, \boldsymbol{\phi}_0$.
2: **for** $t = 0, \ldots, T - 1$ **do**
3:     Send $\boldsymbol{\omega}_t, \boldsymbol{\theta}_t, \boldsymbol{\phi}_t, D_p$ (optional) to all clients.
4:     **for** chosen client $i = 1, \ldots, M$ **do**
5:         $\boldsymbol{\omega}_i^0 = \boldsymbol{\omega}_t, \boldsymbol{\theta}_i^0 = \boldsymbol{\theta}_t, \boldsymbol{\phi}_i^0 = \boldsymbol{\phi}_t, \boldsymbol{\phi}_g = \boldsymbol{\phi}_t$
6:         **for** $k = 1, \ldots, K$ **do**
7:             # Max Step
8:             $\boldsymbol{\theta}_i^k = \boldsymbol{\theta}_i^{k-1} + \eta \nabla_{\boldsymbol{\theta}} \mathcal{L}_{adv}$.
9:             # Min Step
10:            $\boldsymbol{\omega}_i^k = \boldsymbol{\omega}_i^{k-1} - \eta \nabla_{\boldsymbol{\omega}} \mathcal{L}_k$.
11:            $\boldsymbol{\phi}_i^k = \boldsymbol{\phi}_i^{k-1} - \eta \nabla_{\boldsymbol{\phi}} \mathcal{L}_{gen}$.
12:         Send $\boldsymbol{\omega}_i^K, \boldsymbol{\theta}_i^K, \boldsymbol{\phi}_i^K$ to server.
13:     $\boldsymbol{\omega}_{t+1} = \frac{1}{M} \sum_{i=1}^{M} \boldsymbol{\omega}_i^K$.
14:     $\boldsymbol{\theta}_{t+1} = \frac{1}{M} \sum_{i=1}^{M} \boldsymbol{\theta}_i^K$.
15:     $\boldsymbol{\phi}_{t+1} = \frac{1}{M} \sum_{i=1}^{M} \boldsymbol{\phi}_i^K$.

Algorithm 1: Algorithm Framework of FedDebias

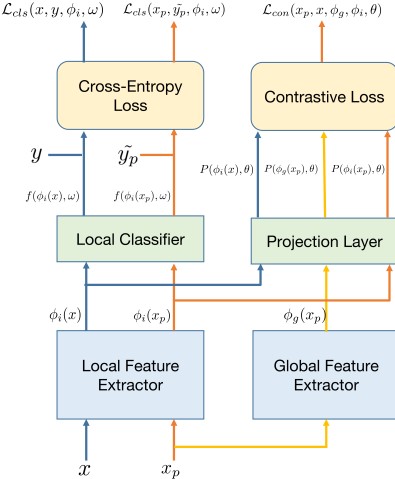

Figure 4: **Overview of FedDebias.** We illustrate how to calculate the three terms in Equation (2) and (3). We calculate the cross-entropy loss of local data $(\mathbf{x}, \mathbf{y})$, and pseudo-data $(\mathbf{x}_p, \tilde{\mathbf{y}}_p)$, and use the local feature $\boldsymbol{\phi}_i(\mathbf{x}), \boldsymbol{\phi}_i(\mathbf{x}_p)$, and global feature $\boldsymbol{\phi}_g(\mathbf{x}_p)$ for contrastive loss.

- **Mixture of local samples and the RSM of a proxy dataset (Mixture).** This strategy relies on applying the procedure of RSM to an irrelevant and globally shared proxy data (refer to Algorithm 3). To guard the distribution distance between the pseudo-data and local data, one sample of the pseudo-data at each client is constructed by

$$\tilde{\mathbf{x}}_p = \frac{1}{K+1} \left( \mathbf{x}_p + \sum_{k=1}^{K} \mathbf{x}_k \right), \qquad \tilde{\mathbf{y}}_p = \frac{1}{K+1} \left( \frac{1}{C} \cdot \mathbf{1} + \sum_{k=1}^{K} \mathbf{y}_k \right), \qquad (4)$$

where $\mathbf{x}_p$ is one RSM sample of the global proxy dataset, and $\mathbf{x}_k$ and $\mathbf{y}_k$ correspond to the data and label of one local sample (vary depending on the client). $K$ is a constant that controls the closeness between the distribution of pseudo-data and local data. As we will show in Section 5, setting $K = 1$ is data efficient yet sufficient to achieve good results.

### 4.3 COMPONENT 1: REDUCING BIAS IN LOCAL CLASSIFIERS

Due to the issue of label distribution skew or the absence of some samples for the majority/minority classes, the trained local model classifier tends to overfit the locally presented classes, and may further hinder the quality of feature extractor (as justified in Figure 3 and Definition 3.3).

As a remedy, here we implicitly mimic the global data distribution—by using the pseudo-data constructed in Section 4.2—to regularize the outputs and thus debias the classifier:

$$\lambda \mathbb{E}_{\mathbf{x}_p \sim D_i} \left[ \mathcal{L}_{cls}(\mathbf{x}_p, \tilde{\mathbf{y}}_p, \boldsymbol{\phi}_i, \boldsymbol{\omega}) \right].$$

Note that the Component 1 is appeared to be the second term of Equation (3).

### 4.4 COMPONENT 2: REDUCING BIAS IN LOCAL FEATURES

In addition to alleviating the biased local classifier in Section 4.3, here we introduce a crucial adversary strategy to learn unbiased local features.

**Intuition of constructing an adversarial problem.** As discussed in Definition 3.3, effective federated learning on heterogeneous data requires learning debiased local feature extractors that 1) can extract local features that are close to global features of the same input data; 2) can extract different local features for input samples from different distributions. However, existing methods that directly minimize the distance between global features and local features (Li et al., 2018b; 2021) have limited performance gain (c.f. Table 1) due to the diminishing optimization objective caused by the indistinguishability between the global and local features of the same input. To this end, we propose to extend the idea of adversarial training to our FL scenarios:

1. We construct a projection layer as the critical step to distinguish features extracted by the global and local feature extractor: such layer ensures that the projected features extracted by local feature extractor will be close to each other (even for distinct local data distributions), but the difference between features extracted by the global and local feature extractor after projection will be considerable (even for the same input samples).

2. We can find that constructing such a projection layer can be achieved by maximizing the local feature bias discussed in Definition 3.3. More precisely, it can be achieved by maximizing the distance between global and local features of pseudo-data and simultaneously minimizing the distance between local features of pseudo-data and local data.

3. We then minimize the local feature biases (discussed in Definition 3.3) under the trained projection space, so as to enforce the learned local features of pseudo-data to be closer to the global features of pseudo-data but far away from the local features of real local data.

**On the importance of utilizing the projection layer to construct the adversary problem.** To construct the aforementioned adversarial training strategy, we consider using an additional projection layer to map features onto the projection space[5]. In contrast to the existing works that similarly add a projection layer (Li et al., 2021), we show that 1) simply adding a projection layer as part of the feature extractor has trivial performance gain (c.f. Figure 5(a)); 2) our design is the key step to reducing the feature bias and boosting the federated learning on heterogeneous data (c.f. Table 3).

**Objective function design.** We extend the idea of Li et al. (2021) and improve the contrastive loss initially proposed in simCLR (Chen et al., 2020) to our challenging scenario. Different from previous works, we use the projected features (global and local) on pseudo-data as the positive pairs and rely on the projected local feature of both pseudo-data and local data as the negative pairs:

$$\mathcal{L}_{con}(\mathbf{x}_p, \mathbf{x}, \phi_g, \phi_i, \boldsymbol{\theta}) = -\log\left(\frac{\exp\left(\frac{\text{sim}\left(P(\phi_i(\mathbf{x}_p)), P(\phi_g(\mathbf{x}_p))\right)}{\tau_1}\right)}{\exp\left(\frac{\text{sim}\left(P(\phi_i(\mathbf{x}_p)), P(\phi_g(\mathbf{x}_p))\right)}{\tau_1}\right) + \exp\left(\frac{\text{sim}\left(P(\phi_i(\mathbf{x}_p)), P(\phi_i(\mathbf{x}))\right)}{\tau_2}\right)}\right), \quad (5)$$

where $P$ is the projection layer parameterized by $\boldsymbol{\theta}$, $\tau_1$ and $\tau_2$ are temperature parameters, and sim is the cos-similarity function. Our implementation uses a tied value for $\tau_1$ and $\tau_2$ for the sake of simplicity, but an improved performance may be observed by tuning these two.

## 5 EXPERIMENTS

### 5.1 EXPERIMENT SETTING

We elaborate the detailed experiment settings in Appendix A.

**Baseline algorithms.** We compare FedDebias with both FL baselines and commonly used domain generalization (DG) baselines that can be adapted to FL scenarios. Note that we do not consider domain generalization scenarios and include DG baselines to check if DG methods can benefit FL on non-iid clients. For FL baselines, we choose FedAvg (McMahan et al., 2016), Moon (Li et al., 2021), FedProx (Li et al., 2018b), VHL (Tang et al., 2022), and FedMix (Yoon et al., 2021b), which are most relevant to our proposed algorithms. For DG baselines, we choose GroupDRO (Sagawa et al., 2019), Mixup (Yan et al., 2020), and DANN (Ganin et al., 2015). Unless specially mentioned, all algorithms use FedAvg as the backbone algorithm.

**Models and datasets.** We examine all algorithms on RotatedMNIST, CIFAR10, and CIFAR100 datasets. We use a four-layer CNN for RotatedMNIST, VGG11 for CIFAR10, and Compact Convolutional Transformer (CCT (Hassani et al., 2021)) for CIFAR100. We split the datasets following the idea introduced in Yurochkin et al. (2019); Hsu et al. (2019); Reddi et al. (2021), where we leverage the Latent Dirichlet Allocation (LDA) to control the distribution drift with parameter $\alpha$. The pseudo-data is chosen as **RSM** by default, and we also provide results on other types of pseudo-data (c.f. Figure 5(c)). By default, we generate one batch of pseudo-data (64 for MNIST and 32 for other datasets) in each round, and we also investigate only generating one batch of pseudo-data at the beginning of training to reduce the communication cost (c.f. Figure 5(b), Figure 5(c)). We use SGD optimizer (with momentum=0.9 for CCT), and set the learning rate to 0.001 for RotatedMNIST, and 0.01 for other datasets. The local batch size is set to 64 for RotatedMNIST, and 32 for other datasets (following the default setting in DomainBed (Gulrajani & Lopez-Paz, 2020)). Additional

---

[5]Such a projection layer is not part of the feature extractor or used for classification, as shown in Figure 4.

Table 1: **Performance of algorithms.** We split RotatedMNIST, CIFAR10, and CIFAR100 to 10 clients with $\alpha = 0.1$, and ran 1000 communication rounds on RotatedMNIST and CIFAR10 for each algorithm, 800 communication rounds CIFAR100. We report the mean of maximum (over rounds) 5 test accuracies and the number of communication rounds to reach the threshold accuracy.

| Algorithm | RotatedMNIST (CNN) | | CIFAR10 (VGG11) | | CIFAR100 (CCT) | |
|---|---|---|---|---|---|---|
| | Acc (%) | Rounds for 80% | Acc (%) | Rounds for 55% | Acc (%) | Rounds for 43% |
| FedAvg | 82.47 | 828 (1.0X) | 58.99 | 736 (1.0X) | 44.00 | 550 (1.0X) |
| FedProx | 82.32 | 824 (1.0X) | 59.14 | 738 (1.0X) | 43.09 | 756 (0.7X) |
| Moon | 82.68 | 864 (0.9X) | 58.23 | 820 (0.9X) | 42.87 | 766 (0.7X) |
| DANN | 84.83 | 743 (1.1X) | 58.29 | 782 (0.9X) | 41.83 | - |
| GroupDRO | 80.23 | 910 (0.9X) | 56.57 | 835 (0.9X) | 44.34 | 444 (1.2X) |
| FedDebias (Ours) | **86.58** | **628 (1.3X)** | 64.65 | 496 (1.5X) | 45.14 | 352 (1.5X) |
| FedAvg + Mixup | 82.56 | 840 (1.0X) | 58.57 | 826 (0.9X) | 46.37 | 358 (1.6X) |
| FedMix | 81.33 | 902 (0.9X) | 57.37 | 872 (0.8X) | 42.69 | - |
| FedDebias + Mixup (Ours) | 83.42 | 736 (1.1X) | **65.32** | **392 (1.9X)** | **47.75** | 294 (1.9X) |

Table 2: **Comparison with VHL.** We split CIFAR10 and CIFAR100 to 10 clients with $\alpha = 0.1$, and run 1000 communication rounds on CIFAR10 for each algorithm and 800 communication rounds on CIFAR100. We report the mean of maximum (over rounds) 5 test accuracies and the number of communication rounds to reach the threshold accuracy. We set different numbers of virtual data to check the performance of VHL, and pseudo-data only transfer once in FedDebias (32 pseudo-data). For CIFAR100, we choose Mixup as the backbone.

| Algorithm | CIFAR10 (VGG11) | | CIFAR100 (CCT) | |
|---|---|---|---|---|
| | Acc (%) | Rounds for 60% | Acc (%) | Rounds for 46% |
| VHL (2000 virtual data) | 61.23 | 886 (1.0X) | 46.80 | 630 (1.0X) |
| VHL (20000 virtual data) | 59.65 | 998 (0.9X) | 46.51 | 714 (0.9X) |
| FedDebias (32 pseudo-data) | **64.61** | **530 (1.8X)** | **47.67** | **554 (1.1X)** |

Table 3: **Ablation studies of FedDebias** on the effects of two components. We show the performance of two components, and remove the max step (Line 8 in Algorithm 1) of component 2. We split RotatedMNIST, CIFAR10, and CIFAR100 to 10 clients with $\alpha = 0.1$. We run 1000 communication rounds on RotatedMNIST and CIFAR10 for each algorithm and 800 communication rounds on CIFAR100. We report the mean of maximum (over rounds) 5 test accuracies and the number of communication rounds to reach the target accuracy.

| Algorithm | RotatedMNIST (CNN) | | CIFAR10 (VGG11) | | CIFAR100 (CCT) | |
|---|---|---|---|---|---|---|
| | Acc (%) | Rounds for 80% | Acc (%) | Rounds for 55% | Acc (%) | Rounds for 43% |
| FedAvg | 82.47 | 828 (1.0X) | 58.99 | 736 (1.0X) | 46.37 | 358 (1.0X) |
| Component 1 | 84.40 | 770 (1.1X) | 64.32 | **442 (1.7X)** | 47.22 | 330 (1.1X) |
| Component 2 | 86.25 | 648 (1.3X) | 63.44 | 483 (1.5X) | **47.78** | 308 (1.2X) |
| + w/o max step | 81.24 | 926 (0.9X) | 58.84 | 584 (1.3X) | 43.50 | 512 (0.7X) |
| FedDebias | **86.58** | **628 (1.3X)** | **64.65** | 496 (1.5X) | **47.75** | **294 (1.2X)** |

results regarding the impact of hyper-parameter choices and performance gain of FedDebias on other datasets/settings/evaluation metrics can be found in Appendix C.

## 5.2 NUMERICAL RESULTS

**The superior performance of FedDebias over existing FL and DG algorithms.**[6] In Table 1, we show the results of baseline methods as well as our proposed FedDebias algorithm. When comparing different FL and DG algorithms, we discovered that: 1) FedDebias performs best in all settings; 2) DG baselines only slightly outperform ERM, and some are even worse; 3) Regularizing local models to global models from prior rounds, such as Moon and Fedprox, does not result in positive outcomes.

**Comparison with VHL.** We vary the size of virtual data in VHL and compare it with our FedDebias in Table 2: our communication-efficient FedDebias only uses 32 pseudo-data and transfers pseudo-data once, while the communication-intensive VHL (Tang et al., 2022) requires the size of virtual data to be proportional to the number of classes and uses at least 2,000 virtual data (the authors suggest 2,000 for CIFAR10 and 20,000 for CIFAR100 respectively in the released official code, and we use the default value of hyper-parameters and implementation provided by the authors). We can find that 1) FedDebias always outperforms VHL. 2) FedDebias overcomes several shortcomings of VHL, e.g., the need for labeled virtual data and the large size of the virtual dataset.

---

[6]We give the results of CIFAR10 with ResNet18 in Table 6 of Appendix C.

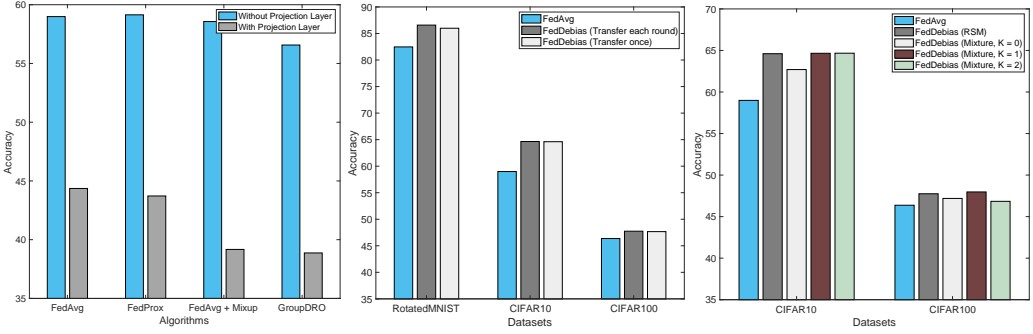

(a) w/ and w/o projection layer.   (b) communication of pseudo-data.   (c) choices of pseudo-data.

Figure 5: **Ablation studies of FedDebias**, regarding the impact of projection layer, the communication strategy of pseudo-data, and the choices of pseudo-data. In Figure 5(a), we show the performance of algorithms with/without the additional projection layer on CIFAR10 dataset with the VGG11 model. In Figure 5(b), we show the performance of FedDebias on RotatedMNIST, CIFAR10, and CIFAR100 datasets when only transferring pseudo-data once (at the beginning of training) or generating new pseudo-data each round. In Figure 5(c), we show the performance of FedDebias using different types of pseudo-data (all transfer once at the beginning of training). We split each dataset into 10 clients with $\alpha = 0.1$ and used CNN for RotatedMNIST dataset, VGG11 for CIFAR10, and CCT for CIFAR100. We run 1000 communication rounds on RotatedMNIST and CIFAR10 for each algorithm and 800 communication rounds on CIFAR100. We report the mean of maximum 5 test accuracies.

Table 4: **Performance of FedDebias on CIFAR10 with different number of clients.** We split CIFAR10 dataset into 10, 30, and 100 clients with $\alpha = 0.1$. We run 1000 communication rounds for each algorithm on the VGG11 model, and report the mean of maximum 5 accuracies (over rounds) during training on test datasets.

| Methods | Acc (%) with 10 clients | Acc (%) with 30 clients | Acc (%) with 100 clients |
|---|---|---|---|
| FedAvg | 58.99 | 44.83 | 38.20 |
| FedDebias | **64.65** | **50.28** | **41.59** |

## 5.3 ABLATION STUDIES

**Effectiveness of the different components in FedDebias.**    In Table 3, we show the improvements brought by different components of FedDebias. In order to highlight the importance of our two components, especially the max-step (c.f. Line 8 in Algorithm 1) in component 2, we first consider two components of FedDebias individually, followed by removing the max-step. We find that: 1) Two components of FedDebias have individual improvements compared with FedAvg, but the combined solution FedDebias consistently achieves the best performance. 2) The projection layer is crucial. After removing projection layers, the component 2 of FedDebias performs even worse than FedAvg; such insights may also explain the limitations of Moon (Li et al., 2021).

**Performance of FedDebias on CIFAR10 with different number of clients.**    In Table 4, we vary the number of clients among $\{10, 30, 100\}$. For each setting, 10 clients are randomly chosen in each communication round. FedDebias outperforms FedAvg by a significant margin in all settings.

**Reducing the communication cost of FedDebias.**    To reduce the communication overhead, we reduce the size of pseudo-data, and only transmit one mini-batch of pseudo-data (64 for MNIST and 32 for others) once at the beginning of training. In Figure 5(b), we show the performance of FedDebias when pseudo-data only transfer to clients at the beginning of the training (64 pseudo-data for RotatedMNIST, and 32 for CIFAR10 and CIFAR100). Results show that only transferring pseudo-data once can achieve comparable performance gain compared with transferring pseudo-data in each round. This indicates that the performance of FedDebias will not drop even we give a small number of pseudo-data.

**Regarding privacy issues caused by RSM.**    Because RSM may have some privacy issues, we consider using Mixture to protect privacy. In Figure 5(c), we show the performance of FedDebias with different types of pseudo-data (pseudo-data only transfer once at the beginning of training as in Figure 5(b)). Results show that: 1) FedDebias consistently outperforms FedAvg on all types of pseudo-data. 2) When using **Mixture** as pseudo-data and setting $K = 0$ (Equation (4)), FedDebias still have a performance gain compared with FedAvg, and a more significant performance gain can be observed by setting $K = 1$.

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

# A    EXPERIMENT DETAILS

**Framework and baseline algorithms.**    In addition to traditional FL methods, we aim to see if domain generalization (DG) methods can help increase model performance during FL training. Thus, we use the DomainBed benchmark (Gulrajani & Lopez-Paz, 2020), which contains a series of regularly used DG algorithms and datasets. The algorithms in DomainBed can be divided into three categories:

- **Infeasible methods:** Some algorithms can't be applied in FL scenarios due to the privacy concerns, for example, MLDG (Li et al., 2017), MMD (Li et al., 2018a), CORAL (Sun & Saenko, 2016), VREx (Krueger et al., 2020) that need features or data from each domain in each iteration.
- **Feasible methods (with limitations):** Some algorithms can be applied in FL scenarios with some limitations. For example, DANN (Ganin et al., 2015), CDANN (Li et al., 2018c) require knowing the number of domains/clients, which is impractical in the cross-device setting.
- **Feasible methods ( without limitations):** Some algorithms can be directly applied in FL settings. For example, ERM, GroupDRO (Sagawa et al., 2019), Mixup (Yan et al., 2020), and IRM (Arjovsky et al., 2019).

We choose several common used DG algorithms that can easily be applied in Fl scenarios, including ERM, GroupDRO (Sagawa et al., 2019), Mixup (Yan et al., 2020), and DANN (Ganin et al., 2015). For FL baselines, we choose FedAvg (McMahan et al., 2016) (equal to ERM), Moon (Li et al., 2021), FedProx (Li et al., 2018b), SCAFFOLD (Karimireddy et al., 2019) and FedMix (Yoon et al., 2021b) which are most related to our proposed algorithms.

Notice that some existing works consider combining FL and domain generalization. For example, combining DRO with FL (Mohri et al., 2019; Deng et al., 2021), and combine MMD or DANN with FL (Peng et al., 2019; Wang et al., 2022; Shen et al., 2021). The natural idea of the former two DRO-based approaches is the same as our GroupDRO implementations, with some minor weight updates differences; the target of the later series of works that combine MMD or DANN is to train models to work well on unseen distributions, which is orthogonal with our consideration (overcome the local heterogeneity).To check the performance of this series of works, we choose to integrate FL and DANN into our environments.

Notice that we carefully tune all the baseline methods. The implementation detail of each algorithm is listed below:

- GroupDRO: The weight of each client is updated by $\boldsymbol{\omega}_i^{t+1} = \boldsymbol{\omega}_i^t \exp(0.01 l_i^t)$, where $l_i^t$ is the loss value of client $i$ at round $t$.
- Mixup: Local data is mixed by $\tilde{\mathbf{x}} = \lambda \mathbf{x}_i + (1 - \lambda)\mathbf{x}_j$, and $\lambda$ is sampled by $Beta(0.2, 0.2)$.
- DANN: Use a three-layer MLP as domain discriminator, where the width of MLP is 256. The weight of domain discriminate loss is tuned in $\{0.01, 0.1, 1\}$.
- FedProx: The weight of proximal term is tuned in $\{0.001, 0.01, 0.1\}$.
- Moon: The projection layer is a two-layer MLP, the MLP width is setting to 256, and the output dimension is 128. We tuned the weight of contrastive loss in $\{0.01, 0.1, 1, 10\}$.
- FedMix: The mixup weight $\lambda$ used in FedMix is tuned in $\{0.01, 0.1, 0.2\}$, we construct 64 augmentation data in each local step for RotatedMNIST, and 32 samples for CIFAR10 and CIFAR100..
- VHL: We use the same setting as in the original paper, with the weight of augmentation classification loss $\alpha = 1.0$, and use the "proxy_align_loss" provided by the authors for feature alignment. Virtual data is generated by untrained style-GAN-v2, and we sample 2000 virtual data for CIFAR10 and RotatedMNIST; 20000 virtual data for CIFAR100 follow the default setting of the original work. To make a fair comparison, we sample 32 virtual samples in each local step for CIFAR10 and CIFAR100.
- FedDebias: We use a three-layer MLP as the projection layer, the MLP width is set to 256, and the output dimension is 128. By default, we set $\tau_1 = \tau_2 = 2.0$, the weight of contrastive loss $\mu = 0.5$, and the weight of AugMean $\lambda = 1.0$ on MNIST and CIFAR100, $\lambda = 0.1$ on CIFAR10 and PACS. We sample 64 pseudo-data in each local step for RotatedMNIST and 32 samples for CIFAR10 and CIFAR100.

**Feature correction when using proxy datasets to construct pseudo-data.**    When using proxy datasets to construct the pseudo-data, we additionally mix up local data with pseudo-data to make the pseudo-data not too far from the local distribution. However, the pseudo-data will have a large overlap with local data after the mixup. Then the $\exp\left(\frac{\text{sim}(P(\phi_i(x_p)), P(\phi_i(x)))}{\tau_2}\right)$ term in Equation (5),

which is used to maximize the distance between local features of local data and pseudo-data, will be meaningless. To address this issue, we change this term to

$$\exp\left(\frac{\text{sim}\left(P(\phi_i(\mathbf{x}_p) - \langle \tilde{\mathbf{y}}_p, \mathbf{y}\rangle \cdot \phi_i(\mathbf{x})), P(\phi_i(\mathbf{x}))\right)}{\tau_2}\right), \tag{6}$$

where $\tilde{y}_p$ is the pseudo-label of $x_p$, and $y$ is the one-hot label of local data $x$. Then we can minimize the relationship between $x$ and $x_p$ caused by the mixup with local data.

**Datasets and Models.** For datasets, we choose RotatedMNIST, CIFAR10, CIFAR100, and PACS. For RotatedMNIST, CIFAR10, and CIFAR100, we split the datasets following the idea introduced in Yurochkin et al. (2019); Hsu et al. (2019); Reddi et al. (2021), where we leverage the Latent Dirichlet Allocation (LDA) to control the distribution drift with parameter $\alpha$. Larger $\alpha$ indicates smaller non-iidness. We divided each environment into two clients for PACS, with the first client containing data from classes 0-3, and the second client containing data from classes 4-6.

Unless specially mentioned, we split RotatedMNIST, CIFAR10, and CIFAR100 to 10 clients and set $\alpha = 0.1$. For PACS, we have 8 clients instead. Notice that for each client of CIFAR10, we utilize a special transformation, i.e., rotation to the local data, to simulate the natural shift. In detail:

- RotatedMNIST: We first split MNIST by LDA using parameter $\alpha = 0.1$ to 10 clients, then for each client, we rotate the local data by $\{0, 15, 30, 45, 60, 75, 90, 105, 120, 135\}$.
- CIFAR10: We first split CIFAR10 by LDA using parameter $\alpha = 0.1$ to $N$ clients. Then for each client, we sample $q \in \mathbb{R}^{10}$ from $Dir(1.0)$. For each image in local data, we sample an angle in $\{0, 15, 30, 45, 60, 75, 90, 105, 120, 135\}$ by probability $q$, and rotate the image by the angle.
- Clean CIFAR10: Unlike the previous setting, we do not rotate the samples in CIFAR10 (no inner-class non-iidness).
- CIFAR100: We split the CIFAR100 by LDA using parameter $\alpha = 0.1$, and transform the train data using RandomCrop, RandomHorizontalFlip, and normalization.

Each communication round includes 50 local iterations, with 1000 communication rounds for RotatedMNIST and CIFAR10, 800 communication rounds for CIFAR100, and 400 communication rounds for PACS. Notice that the number of communication rounds is carefully chosen, and the accuracy of all algorithms does not significantly improve after the given communication rounds.

The public data is chosen as RSM (Yoon et al., 2021b) by default, and we also provide results on other proxy datasets. We utilize a four-layer CNN for MNIST, VGG11 for CIFAR10 and PACS, and CCT (Hassani et al., 2021) (Compact Convolutional Transformer, cct_7_3x1_32_c100) for CIFAR100.

For each algorithm and dataset, we employ SGD as the optimizer, and set learning rate $lr = 0.001$ for MNIST, and $lr = 0.01$ for CIFAR10 , CIFAR100, and PACS. When using CCT and ResNet, we set momentum as 0.9. We set the same random seeds for all algorithms. We set local batch size to 64 for RotatedMNIST, and 32 for CIFAR10, CIFAR100, and PACS.

## B    DETAILS OF AUGMENTATION DATA

We use the data augmentation framework the same as FedMix, as shown in Algorithm 2. For each local dataset, we upload the mean of each $M$ samples to the server. The constructed augmentation data is close to random noise. As shown in Figure 6, we randomly choose one sample in the augmentation dataset of CIFAR10 dataset.

---

**Algorithm 2** Construct Augmentation Data

---

**Require:** local Datasets $D_1, \ldots, D_N$, number of augmentation data for each client $K$, number of samples to construct one augmentation sample $M$.
**Ensure:** Augmentation Dataset $D_p$.
 1: Initialize $D_p = \emptyset$.
 2: **for** $i = 1, \ldots, N$ **do**
 3:     **for** $k = 1, \ldots, K$ **do**
 4:         Randomly sample $x_1, \ldots, x_M$ from $D_i$.
 5:         $\bar{x} = \frac{1}{M}\sum_{m=1}^{M} x_M$.
 6:         $D_p = D_p \cup \{\bar{x}\}$

---

---

**Algorithm 3** Construct Augmentation Data by Proxy Data

---

**Require:** Proxy Datasets $D_{prox}$, number of augmentation data $K$, number of samples to construct one augmentation sample $M$.

**Ensure:** Augmentation Dataset $D_p$.

1: Initialize $D_p = \emptyset$.
2: **for** $k = 1, \ldots, K$ **do**
3:     Randomly sample $x_1, \ldots, x_M$ from $D_{prox}$.
4:     $\bar{x} = \frac{1}{M} \sum_{m=1}^{M} x_M$.
5:     $D_p = D_p \cup \{\bar{x}\}$

---

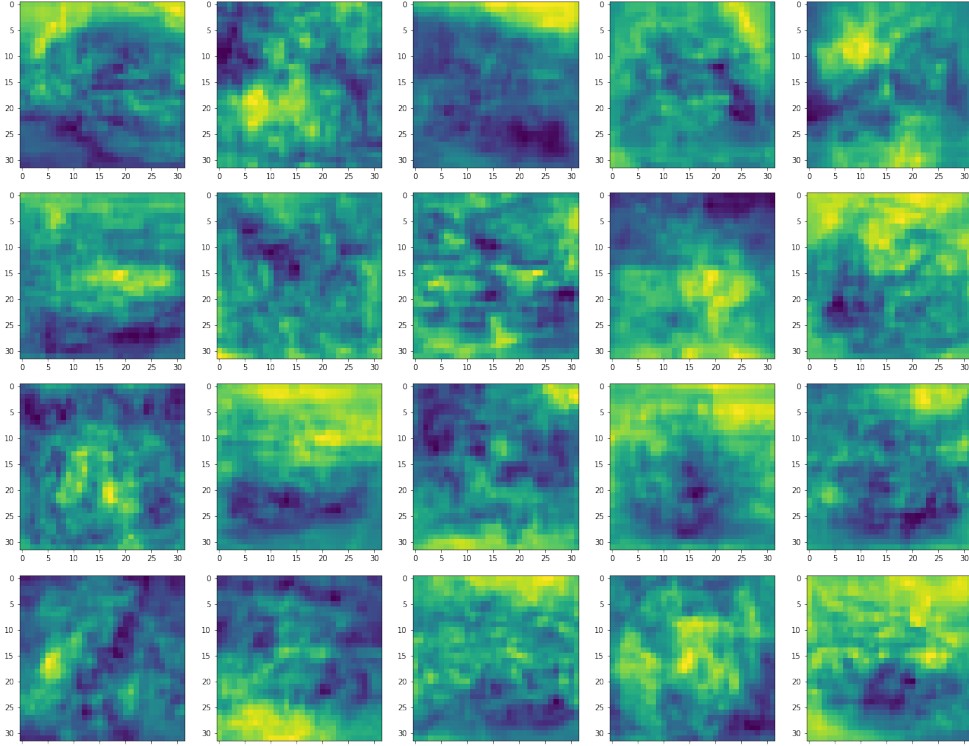

Figure 6: We show 20 augmentation data of CIFAR10 dataset here. Notice that the augmentation data is close to random noise and can not be classified as any class.

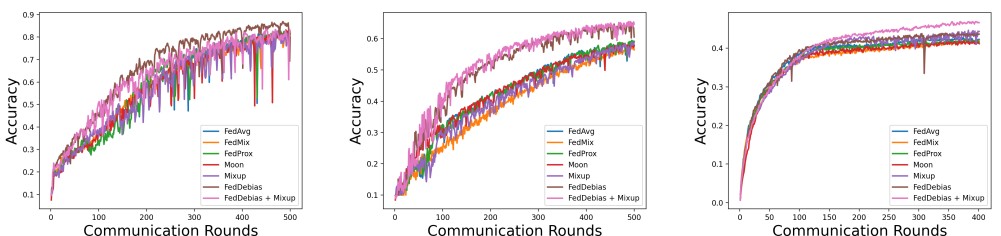

(a) Convergence curve on RotatedMNIST    (b) Convergence curve on CIFAR10    (c) Convergence curve on CIFAR100

Figure 7: Convergence curve of algorithms on different datasets.

## C  ADDITIONAL RESULTS

### C.1  RESULTS WITH ERROR BAR

In this section, we report the performance of our method FedAug and other baselines with an error bar to verify the performance gain of our proposed method.

Table 5: **Performance of algorithms with error bar.** All examined algorithms use FedAvg as the backbone. We run 1000 communication rounds on RotatedMNIST and CIFAR10 for each algorithm. For each algorithm, we run three different trials with different random seeds. For each trial, we report the mean of maximum 5 accuracies for test datasets and the number of communication rounds to reach the threshold accuracy.

| Algorithm | RotatedMNIST | | CIFAR10 | |
|---|---|---|---|---|
| | Acc (%) | Rounds for 80% | Acc (%) | Rounds for 55% |
| ERM (FedAvg) | $82.78 \pm 0.38$ | 821 (1.0X) | $58.97 \pm 0.30$ | 742 (1.0X) |
| DANN | $84.67 \pm 0.46$ | 754 (1.1X) | $58.98 \pm 0.61$ | 747 (1.0X) |
| Mixup | $82.38 \pm 0.07$ | 853 (1.0X) | $58.32 \pm 0.33$ | 822 (0.9X) |
| GroupDRO | $80.65 \pm 0.53$ | 929 (0.9X) | $56.72 \pm 0.26$ | 840 (0.9X) |
| FedDebias (Ours) | $\mathbf{87.05 \pm 0.44}$ | **637 (1.3X)** | $\mathbf{64.62 \pm 0.32}$ | **374 (2.0X)** |

Table 6: **Performance of algorithms on CIFAR10.** We split CIFAR10 dataset to 10 clients with $\alpha = 0.1$, without additional rotation. For each algorithm, we run 1000 communication rounds on ResNet18 (with group normalization), and set local steps to 50. Note that we set momentum to 0.9 for ResNet18.

| | FedAvg | FedProx | Moon | VHL | FedDebias (ours) |
|---|---|---|---|---|---|
| Accuracy (ResNet18) | 45.91 | 46.28 | 43.85 | 43.7 | 47.29 |

### C.2  ABLATION STUDY OF FEDDEBIAS

**Values of $\tau_1$ and $\tau_2$ in Componennt 2.**  In this paragraph, we investigate how the value of $\tau_1$ and $\tau_2$ affect the performance of the second component of FedDebias. In table 7, we show the results on Rotated-MNIST dataset with different weights $\tau_1$ and $\tau_2$. Results show that: 1) Setting $\tau_2 = 0$ , which only minimizes the distance of global and local features, has significant performance gain compare with ERM. However, adding $\tau_2$ can further improve the performance. 2) The best weight on Rotated-MNIST dataset is $\tau_1 = 2.0$ and $\tau_2 = 0.5$.

Table 7: **Performance of Component 2 of FedDebias under different values of $\tau_1, \tau_2$.** We run 1000 communication rounds on RotatedMNIST dataset. For each setting, we run three different trials with different random seeds. For each trial, we report the mean of maximum 5 accuracies for test datasets and the number of communication rounds to reach the threshold accuracy.

| $\tau_1$ | $\tau_2$ | Acc (%) | Rounds for 80% | Rounds for 85% |
|---|---|---|---|---|
| 2.0 | 0.0 | $86.11 \pm 0.77$ | 746 | 933 |
| 2.0 | 0.1 | $86.22 \pm 0.33$ | 753 | 920 |
| 2.0 | 0.5 | $87.24 \pm 0.50$ | 647 | 851 |
| 2.0 | 1.0 | $86.25 \pm 0.87$ | 705 | 922 |
| 2.0 | 2.0 | $86.01 \pm 0.33$ | 680 | 932 |

**Weights of the first component of FedDebias.**  In this paragraph, we investigate how the weights of the first component of FedDebias affect the performance of models in table 9.

**Domain robustness of FL and DG algorithms.**  We also hope that our method can increase the model's robustness because it expects to train client invariant features. Therefore, we calculate the worst accuracy on test datasets of all clients/domains and report the mean of each algorithm's

Table 8: **Performance of FedDebias under different values of** $\tau_1, \tau_2$**.** We run 1000 communication rounds on the CIFAR10 dataset. For each setting, we run three different trials with different random seeds. For each trial, we report the mean of maximum 5 accuracies for test datasets and the number of communication rounds to reach the threshold accuracy.

| $\tau_1$ | $\tau_2$ | Acc (%) | Rounds for 55% | Rounds for 60% |
|------|------|--------------|-----|-----|
| 2.0 | 0.0 | $64.05 \pm 0.27$ | 390 | 563 |
| 2.0 | 0.5 | $64.26 \pm 0.47$ | 382 | 585 |
| 2.0 | 1.0 | $64.77 \pm 0.24$ | 374 | 533 |
| 2.0 | 2.0 | $64.62 \pm 0.32$ | 374 | 541 |

Table 9: **Performance of component 1 under different weights.** We run 1000 communication rounds on the CIFAR10 dataset. For each setting, we run three different trials with different random seeds. For each trial, we report the mean of maximum 5 accuracies for test datasets and the number of communication rounds to reach the threshold accuracy. We use $\lambda$ as the weight of the first component of FedDebias.

| $\lambda$ | Acc (%) | Rounds for 55% | Rounds for 60% |
|------|--------------|-----|-----|
| 0.1 | $64.12 \pm 0.27$ | 442 | 591 |
| 0.5 | $64.92 \pm 0.46$ | 385 | 536 |
| 1.0 | $64.50 \pm 0.34$ | 379 | 565 |

top 5 worst accuracies in Table 11 to show the domain robustness of algorithms. We have the following findings: 1) FedDebias significantly outperforms other approaches, and the improvements of FedDebias over FedAvg become more significant than the mean accuracy in Table 1. FedDebias has a role in learning a domain-invariant feature and improving robustness, as evidenced by this finding. 2) Under these settings, DG baselines outperform FedAvg. This finding demonstrates that the DG algorithms help to enhance domain robustness.

Table 10: **Performance of algorithms.** All examined algorithms use FedAvg as the backbone. We run 1000 communication rounds on RotatedMNIST and CIFAR10 for each algorithm, 800 communication rounds CIFAR100 and 400 communication rounds for PACS. We report the mean of maximum 5 accuracies for test datasets and the number of communication rounds to reach the final accuracy of ERM .

| Algorithm | RotatedMNIST | | CIFAR10 | | PACS | |
|-----------|---------|-------------------|---------|-------------------|---------|-------------------|
|  | Acc (%) | Rounds (Speed up) | Acc (%) | Rounds (Speed up) | Acc (%) | Rounds (Speed up) |
| ERM (FedAvg) | 82.47 | 828 (1.0X) | 58.99 | 736 (1.0X) | 64.03 | 168 (1.0X) |
| FedProx | 82.32 | 824 (1.0X) | 59.14 | 738 (1.0X) | 65.10 | 168 (1.0X) |
| SCAFFOLD | 82.49 | 814 (1.0X) | 59.00 | 738 (1.0X) | 64.49 | 168 (1.0X) |
| FedMix | 81.33 | 902 (0.9X) | 57.37 | 872 (0.8X) | 62.14 | 228 (0.7X) |
| Moon | 82.68 | 864 (0.9X) | 58.23 | 820 (0.9X) | 64.86 | 122 (1.4X) |
| DANN | 84.83 | 743 (1.1X) | 58.29 | 782 (0.9X) | 64.97 | 109 (1.5X) |
| Mixup | 82.56 | 840 (1.0X) | 58.57 | 826 (0.9X) | 64.36 | 210 (0.8X) |
| GroupDRO | 80.23 | 910 (0.9X) | 56.57 | 835 (0.9X) | 64.40 | 170 (1.0X) |
| FedAug (Ours) | **86.58** | **628 (1.3X)** | **64.65** | **496 (1.5X)** | **65.63** | **100 (1.7X)** |

## C.3 T-SNE AND CLASSCIFIER OUTPUT

As the setting in Figure 2 and Figure 3, we investigate if the two components of FedDebias will help for mitigating the proposed bias on feature and classifier. Figure 8 show the features after the second component of FedDebias, which implies this component can significantly mitigate the proposed feature bias: 1) on the seen datasets, local features are close to global features. 2) on the unseen datasets, the local feature is far away from that of seen datasets. Figure 9 shows the output of the local classifier after the first component of FedDebias on unseen classes. Notice that compared with Figure 3, the output is more balanced.

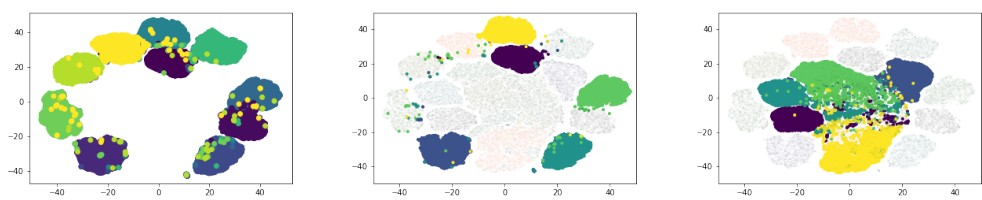

(a) Global feature of $X_1$, $F_g(X_1)$    (b) Local feature of $X_1$, $F_1(X_1)$    (c) Local feature of $X_2$, $F_1(X_2)$

Figure 8: Features after the second component of FedDebias.

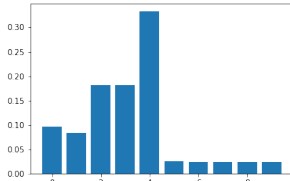

Figure 9: Classifier output after the first component of FedDebias on unseen classes.

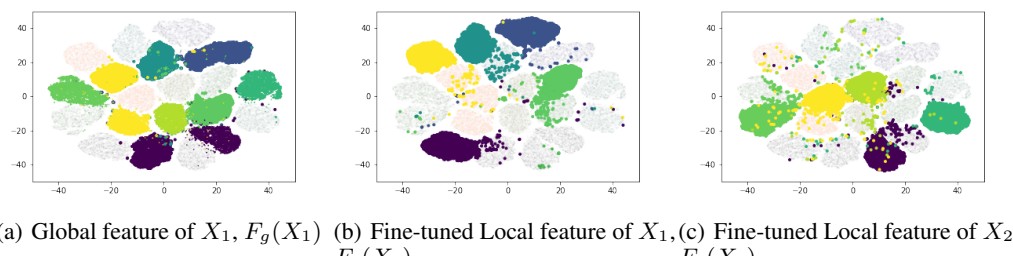

(a) Global feature of $X_1$, $F_g(X_1)$   (b) Fine-tuned Local feature of $X_1$, (c) Fine-tuned Local feature of $X_2$,
                                   $F_1(X_1)$                            $F_1(X_2)$

Figure 10: Fine-tuned local features after 10 local epochs.

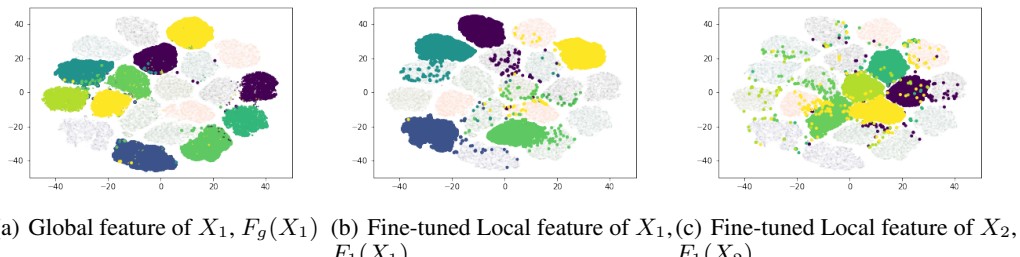

(a) Global feature of $X_1$, $F_g(X_1)$   (b) Fine-tuned Local feature of $X_1$, (c) Fine-tuned Local feature of $X_2$,
                                   $F_1(X_1)$                            $F_1(X_2)$

Figure 11: Fine-tuned local features after 20 local epochs.

Table 11: **Worst Case Performance of algorithms.** All examined algorithms use FedAvg as the backbone. We run 1000 communication rounds on RotatedMNIST and CIFAR10 for each algorithm, 800 rounds for CIFAR100, and 400 communication rounds for PACS. We calculate the worst accuracy for all clients in each round and report the mean of the top 5 worst accuracies for each method. Besides, we report the number of communication rounds to reach the final worst accuracy of FedAvg.

| Algorithm | RotatedMNIST | | CIFAR10 | | PACS | |
|---|---|---|---|---|---|---|
| | Acc (%) | Rounds (Speed up) | Acc (%) | Rounds (Speed up) | Acc (%) | Rounds (Speed up) |
| ERM (FedAvg) | 66.60 | 816 (1.0X) | 41.30 | 846 (1.0X) | 42.79 | 170 (1.0X) |
| FedProx | 65.88 | 780 (1.0X) | 41.84 | 840 (1.0X) | 42.82 | 170 (1.0X) |
| SCAFFOLD | 66.72 | 804 (1.0X) | 40.88 | 840 (1.0X) | 41.63 | 170 (1.0X) |
| FedMix | 60.52 | 910 (0.9X) | 28.44 | - | 38.00 | - |
| Moon | 66.18 | 866 (0.9X) | 40.34 | 908 (0.9X) | 41.59 | 66 (2.6X) |
| DANN | 67.85 | 753 (1.1X) | 43.38 | 747 (1.1X) | 40.51 | 59 (2.9X) |
| Mixup | 66.25 | 836 (1.0X) | 40.32 | 984 (0.9X) | 41.89 | 252 (0.7X) |
| GroupDRO | 68.53 | **568 (1.4X)** | 46.90 | 656 (1.3X) | 43.18 | 246 (0.7X) |
| FedDebias (Ours) | **77.13** | 630 (1.3X) | **48.94** | **632 (1.3X)** | **43.99** | **58 (2.9X)** |

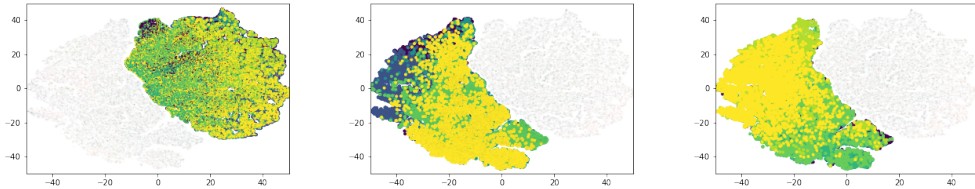

(a) Global feature of $X_1$, $F_g(X_1)$    (b) Fine-tuned Local feature of $X_1$, $F_1(X_1)$    (c) Fine-tuned Local feature of $X_2$, $F_1(X_2)$

Figure 12: First train global model on the whole dataset for 1 epoch, then report local features after 10 local epochs.

In Figure 10 and Figure 11, we show the local learning bias when local model has better feature initialization. We copy the feature extractor of global model to local models, and randomly initialize local classifiers. Results show that: 1) The drifts between global and local features are still significant even has a good feature initialization. 2) The local features of unseen data are less relevant to the local features of seen data compare with training from scratch. This indicates that such a problem will be mitigated after enough training rounds. 3) The drifts between global and local features increase as the number of local epochs increases.

We also investigate if our observation remains for different stages of global models. In this experiment, we use CIFAR10 dataset, and train global model for 1, 3, 10 epochs on the whole dataset to obtain 29.74%, 38.65%, 49.28% global accuracy, then we directly copy global models to clients (including classifier). We fine-tune the global models for 10 local epochs, results are shown in Figure 12, Figure 13 and Figure 14. Results show that: For not well-trained global models, difference between global features on the same input and similarity between local features of different inputs are both significant.

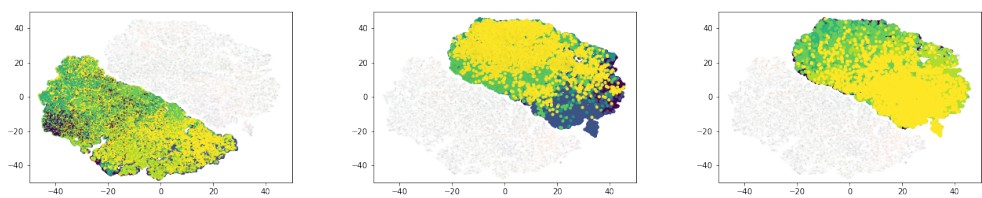

(a) Global feature of $X_1$, $F_g(X_1)$    (b) Fine-tuned Local feature of $X_1$, $F_1(X_1)$    (c) Fine-tuned Local feature of $X_2$, $F_1(X_2)$

Figure 13: First train global model on the whole dataset for 3 epochs, then report local features after 10 local epochs.

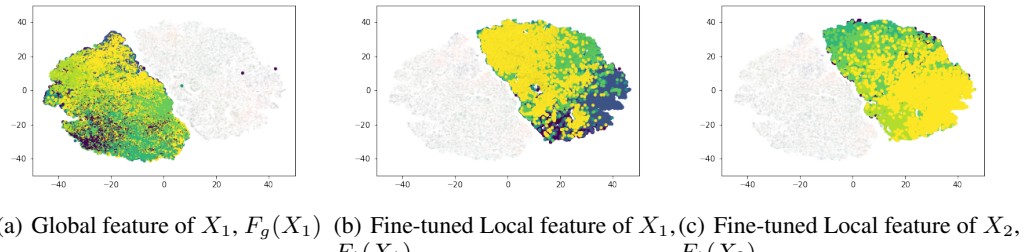

(a) Global feature of $X_1$, $F_g(X_1)$    (b) Fine-tuned Local feature of $X_1$, $F_1(X_1)$    (c) Fine-tuned Local feature of $X_2$, $F_1(X_2)$

Figure 14: First train global model on the whole dataset for 10 epochs, then report local features after 10 local epochs.

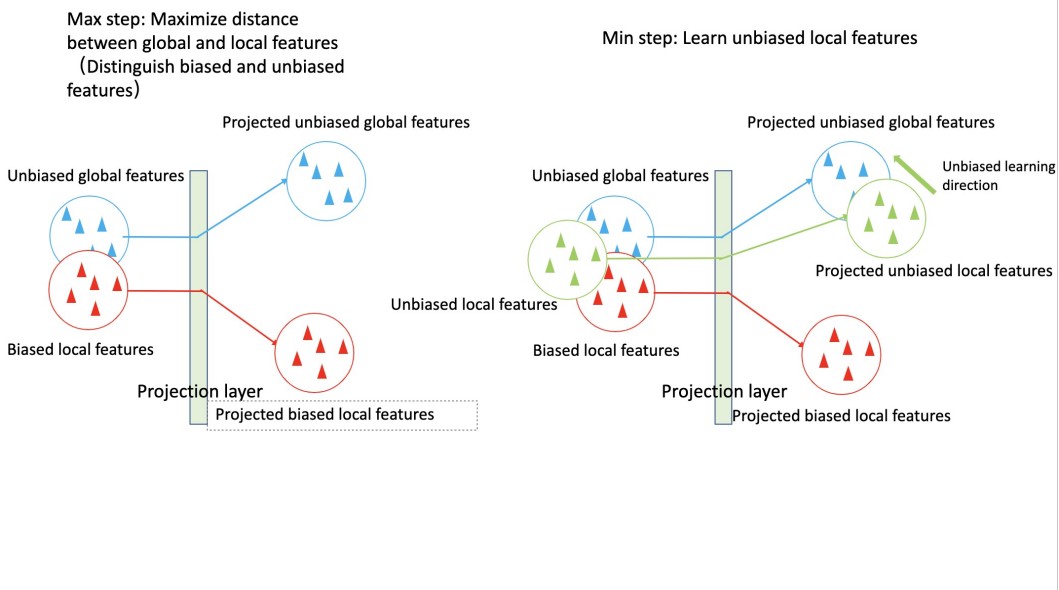

Figure 15: Illustration of the min-max process. In the left side of the figure, we show that the projection layer maximizes the distance between biased and unbiased features. In the right side of the figure, we show that the unbiased local feature is trained by forcing projected features closer to unbiased features.

### C.4 ILLUSTRATION OF MIN-MAX PROBLEM

In Figure 15, we illustrate the intuition to use the proposed min-max process. The projection layer is used to distinguish biased and unbiased features that can not be distinguished well on the original feature space, and the min step is to learn unbiased local features that close to unbiased features on the projected spaces.

