# OpenReview forum: "FedDebias: Reducing the Local Learning Bias Improves Federated Learning on Heterogeneous Data"
_ICLR.cc/2023/Conference — Submitted to ICLR 2023_

### Official Review · Reviewer_cKx8 · 2022-10-23

**Confidence:** 4
**Correctness:** 3
**Technical Novelty And Significance:** 2
**Empirical Novelty And Significance:** 3
**Recommendation:** 5

**Clarity, Quality, Novelty And Reproducibility:**

Clarity: The paper generally presents well.

Quality: Fair. The paper does not provide solid analyses of the proposed approach.

Novelty: The observation of the local bias from the feature view has been shown and investigated in many existing studies (e.g., [1,2]). The min-max process with the pseudo-data is interesting.
[1] Model-contrastive federated learning
[2] Federated Learning for Non-IID Data via Unified Feature Learning and Optimization Objective Alignment

Reproducibility: Good. The authors provide the source code.


**Strength And Weaknesses:**

Strength: The idea is clear. The experiments are comprehensive. The improvement of FedDebias is significant.

Weaknesses:
1. FedDebias needs to transfer pseudo-data during training, which is produced by averaging the raw data. Thus, additional privacy concerns and communication overhead are introduced in FedDebias, which lack analyses and discussion.

2. The paper lacks theoretical analyses of the effectiveness and convergence of the proposed approach.

3. The experiments can be further improved. What is the pseudo-data size generated by RSE? What is the subset size used to construct the samples in RSE? I think the pseudo-data is important in FedDebias. Thus, more investigation and experiments on the pseudo-data (e.g., changing the size of pseudo-data) are needed.

4. I do not understand why it is necessary to maximize the distance between the local features of pseudo-data and local data first. The pseudo-data may have the same class with the local data. I think the paper should provide more insights into why such a min-max process can improve training.

5. Typo: Page 5: maximizing local features -> maximizing the distance between



**Summary Of The Paper:**

The paper proposes FedDebias to address the data heterogeneity issue in federated learning. The paper observes that local features and global features are different even for the same input data and the local models classify all samples into the classes that appear in the local dataset. To address the above biased local features and local classifiers, FedDebias utilizes pseudo-data to improve local training. Specifically, it maximizes the distance between the local features of pseudo-data and local data, and minimizes the distance between the global and local features of pseudo-data. Experiments show that FedDebias outperforms the other FL approaches.

**Summary Of The Review:**

Overall, I think the paper proposes an interesting idea without solid analyses. The paper can be further improved to discuss privacy, effectiveness, and convergence of the proposed approach.

---

> ### Author Response · Authors · 2022-11-11
> **Response to reviewer cKx8 (2/2)**
>
> > ### 3. I do not understand why it is necessary to maximize the distance between the local features of pseudo-data and local data first. The pseudo-data may have the same class as the local data. I think the paper should provide more insights into why such a min-max process can improve training.
>
> Sorry for the misunderstanding. We would like to clarify that *the corresponding discussion was included in sec 4.4; we further added figure (Figure 15) in Appendix C.4 for a clear illustration in the revised paper. The figure is also provided in [this link](https://i.postimg.cc/vZnDVqHn/min-max.jpg).*
>
> - *The min-max process is a key step to make our algorithm design distinct and effective,* as acknowledged by reviewer UkVs with high confidence.
>     - **Without the min-max process, the contrastive loss will gradually diminish.** As observed in Table 3, by removing the min-max process, the contrastive loss will be close to zero after several communication rounds, then the improvement will be trivial.
>     - **The min-max process aids in learning unbiased local features by forcing the distance between local features and unbiased global features on the projection space to be small.** In detail, as we showed in Figure 15 of Appendix C.4,
>         - The distance between biased (local) and unbiased (global) features becomes large after projection, even though they cannot be well distinguished on the original feature space.
>         - The unbiased local feature is learned by forcing local features closer to unbiased features on the projection space.
>
> [4] Li, Qinbin, Bingsheng He, and Dawn Song. "Model-contrastive federated learning." Proceedings of the IEEE/CVF Conference on Computer Vision and Pattern Recognition. 2021.
>
> - **Pseudo-data will not have the same class as local data.** As we showed in Figure 6 of Appendix B, pseudo-data could not be distinguished as any particular class of data, and the unbiased local feature extractor should extract different features for data from different classes.
>
> > ### 4. The paper lacks theoretical analyses of the effectiveness and convergence of the proposed approach.
>
> Thanks for the suggestion. We would like to clarify that different from the existing min-max methods (e.g., DRO [5] and DANN [6]) and contrastive learning methods [7, 8, 9], our method consider different scenarios and objectives:
> - The objective in DRO [5] aims to maximize the worst-case performance on all domains.
> - DANN [6] is designed to learn invariant features and thus improve the model generalization ability on unseen domains.
> - Contrastive learning methods [7, 8, 9] aim to learn features robust to different augmentation methods to improve the model generalization ability.
> - FedDebias instead is designed to reduce the local learning bias in FL with heterogeneous local data so as to improve the scenario where the global training distribution is identical to the test and where all prior methods/analysis cannot apply. Therefore, it is hard to derive a tight theoretical analysis for FedDebias due to the novel scenarios and algorithm designs.
>     - We believe that the insights derived from our design (the local learning bias and min-max approach) and extensive numerical results are significant to the FL community and may inspire further research of theoretical FL.
>
>
> [5] Levy, Daniel, et al. "Large-scale methods for distributionally robust optimization." Advances in Neural Information Processing Systems 33 (2020): 8847-8860.
>
> [6] Ganin, Yaroslav, et al. "Domain-adversarial training of neural networks." The journal of machine learning research 17.1 (2016): 2096-2030.
>
> [7]Chen, Ting, et al. "A simple framework for contrastive learning of visual representations." International conference on machine learning. PMLR, 2020.
>
> [8] Chuang, Ching-Yao, et al. "Debiased contrastive learning." Advances in neural information processing systems 33 (2020): 8765-8775.
>
> [9] Misra, Ishan, and Laurens van der Maaten. "Self-supervised learning of pretext-invariant representations." Proceedings of the IEEE/CVF Conference on Computer Vision and Pattern Recognition. 2020.
>
>
>
> >  ### 5. Page 5: maximizing local features -> maximizing the distance between
>
> Sorry for the typo. We fixed it using blue lines.
>
> ###### We hope the above responses address your concerns. Please let us know if you have other questions. We’re happy to further answer the questions.

---

> ### Author Response · Authors · 2022-11-11
> **Response to reviewer cKx8 (1/2)**
>
> Thanks for your insightful reviews, and we appreciate the valuable suggestions! We've revised the manuscript according to your suggestions. Please kindly find our response to your raised questions below.
>
> > ### 1. FedDebias needs to transfer pseudo-data during training, which is produced by averaging the raw data. Thus, additional privacy concerns and communication overhead are introduced in FedDebias, which lack analyses and discussion.
>
> Thanks for your comment. We also discussed these two problems in Figure 5 (of Section 5.3) in the original submission, and we have enriched the discussions of both problems in Section 5.3 in the revised version.
> - For the communication overhead problem:
>     - **Transmitting pseudo-data will not significantly increase the communication overhead.** As we discussed in  Figure 5(b), we can transmit one mini-batch of pseudo-data to clients only at the beginning of training, and the communication overhead is trivial.
>     - **Existing methods also introduce extra communication overhead** (much more severe than ours). For example, VHL [1] needs to generate virtual data, and the paper uses significantly more virtual data than ours. FedMix [2] needs to transmit pseudo-data generated by `RSM` in each round. Xor Mixup [3] needs to upload encoded data in each round. *Our experiments show that FedDebias can achieve better performance with significantly less communication overhead than the above-listed SOTA methods (Table 1, 2, Figure 5 (b), Figure 5 \(c\))*.
> - For privacy concerns:
>     - *The initial submission has already considered this issue and proposed a `Mixture` (see Sec 4.2) to preserve privacy.*
>          - **The public data collected by the server will not have privacy issues.** All data used in `Mixture` are collected by the server and could be significantly different from local data. Therefore, this operation will not have privacy issues.
>         - **The effectiveness of `Mixture` is justified in Figure 5 \(c\)**: its performance is comparable to and even better than `RSM`, indicating that FedDebias can achieve competitive performance without privacy issues.
>
> [1] Tang, Zhenheng, et al. "Virtual Homogeneity Learning: Defending against Data Heterogeneity in Federated Learning." arXiv preprint arXiv:2206.02465 (2022).
>
> [2] Yoon, Tehrim, et al. "Fedmix: Approximation of mixup under mean augmented federated learning." arXiv preprint arXiv:2107.00233 (2021).
>
> [3] Shin M J, Hwang C, Kim J, et al. Xor mixup: Privacy-preserving data augmentation for one-shot federated learning[J]. arXiv preprint arXiv:2006.05148, 2020.
>
>
>
> > ### 2. What is the pseudo-data size generated by RSE? What is the subset size used to construct the samples in RSE? Need to do experiments on different size of pseudo-data.
>
> Sorry for missing details. We have detailed the number of pseudo-data in the revised paper. Please note that *the experiments on the varying size of pseudo-data were included in the initial submission (Figure 5 (b)).*
> - **One mini-batch (64 for MNIST and 32 for others) of pseudo-data is enough for FedDebias to achieve competitive performance.** As shown in Figure 5(b), we compare the performance of FedDebias when (1) transmitting one mini-batch of pseudo-data in each round and (2) only transmitting one mini-batch of pseudo-data only at the beginning of the training. Results show that the performance of FedDebias does not drop when reducing the number of pseudo-data.
> - As we illustrated in our response to Question 1, the communication overhead of our method is trivial and significantly less than in previous works.
> - **We randomly sample 10 local data from local datasets to construct samples in `RSM`.** (see Algorithm 2 of Appendix B).

---

> ### Author Response · Authors · 2022-11-15
> **Look forward to your feedback!**
>
> Dear reviewer cKx8,
>
> Thank you again for your time and effort in reviewing our paper! As the discussion period is ending soon (November 18th), we would like to know if our responses have addressed your concerns. If they haven't, please let us know what is missing so we can address them. Thank you very much.

---

### Official Review · Reviewer_p1f4 · 2022-10-24

**Confidence:** 3
**Correctness:** 2
**Technical Novelty And Significance:** 2
**Empirical Novelty And Significance:** 2
**Recommendation:** 3

**Clarity, Quality, Novelty And Reproducibility:**

Novelty & Quality:

The idea is quite novel to me but I have reservations concerning its practical feasibility, mainly described in two points below:

+ First, given the pseudo data is devised to be either aggregation of local samples or local samples themselves, I am not sure this method is privacy compliance. In practical setting, sending authentic data to a 3rd party is always a problematic practice; even sending aggregated (but un-sanitized) data to untrusted parties is not allowed because it would still be vulnerable to differential privacy attack and has to be avoided. While I understand that privacy preservation on the algorithmic level is not the focus here but the setting here seem to not even be privacy compliant -- I think the authors should probably discuss this at length to make it clear if this proposal is practically viable.

+ Secondly, how much validation data is needed for the algorithm to work as intended? I imagine such dataset needs to be balanced in terms of label distribution but how do we guarantee that if it has to come from local samples of clients with skewed label distributions? There should also be experiments plotting the effectiveness of the proposed method vs the amount of validation (pseudo) data.

Furthermore, as stated above, there is a lack of coverage & comparison with the existing literature on personalized federated learning, which is meant to address the same problem. In fact, even the prior work of (Yurochkin, 2019) also has a natural mechanism (via probabilistic modeling) that is less affected by the heterogeneous distribution of data across clients. The authors should consider with such method too. Their code is also publicly released.

Clarity:

The paper is mostly well-written but some algorithmic detail appear to be a bit vague & there is -- see my second point in the Quality & Novelty section. In addition, how do we decide the hyper-parameters of the proposed algorithm -- see Eq. (3)

From Eq. (2) and (3), I am curious why phi_g is not part of the learnable parameters? Also, why is x_p \sim D_i in Eq. (3)? Are those typos?

Reproducibility:

The experimental code is released so I believe this work is reproducible.

**Strength And Weaknesses:**

Strengths:

+ The paper is well-written. Key ideas are well-presented.
+ The explanation of the model drift phenomenon under heterogeneous data setup is well put in three main causes all linked to the biased feature representation generated by local learning algorithm when client data are skewed.
+ The key ideas follow quite naturally from the insightful interpretation of model drift.
+ Overall, the motivation is clean & the idea is novel to me.

Weaknesses:

- The technical execution of the idea might not be practically feasible (although the formulation makes sense) -- more on this later
- There is a complete lack of coverage on a very related suite of personalized FL algorithms specifically designed to deal with this issue
- Not clear how much validation data is needed for this to work
- Also, the validation (or pseudo) data obviously needs to be balanced -- how do we ensure that if the design of such validation data comes from random or aggregated sample of local data?

**Summary Of The Paper:**

This paper proposes a new idea of reducing local learning bias to improve FL performance on heterogeneous data. This is achieved via the use of a validation dataset generated as either (a) random samples of local data; or (b) averaged of sampled local data.

The key idea is to solve a min-max optimization task which optimizes simultaneously for (1) a representation mapping that best distinguishes between global and local feature vectors; and (2) local representations and classifiers that produce best (local) performance and also minimize the distance to global representation of the same input.

Intuitively, that means we are seeking local models that produce good local classification performance and have feature representation that cannot be "easily distinguished" from feature representation induced by a model trained on validation data. This means minimizing the performance of a distinguisher that learns to map features to a space on which distance between global & local features of the same input is maximized (hence, min-max optimization).

The proposed algorithm is compared with multiple FL baselines on Rotated MNIST, CIFAR-10 and CIFAR-100 datasets.

**Summary Of The Review:**

This paper introduces a new idea to FL with heterogeneous client data. The idea is new and somewhat interesting but I have concerns regarding its practical feasibility, privacy compliance. I notice that there is also a lack of coverage over a very related FL literature that addresses the same issue. Certain aspects of the proposed algorithm have also not been discussed at necessary length (see my specific comments above).

---

> ### Author Response · Authors · 2022-11-11
> **Response to reviewer p1f4 (3/3)**
>
> > ### 5. given the pseudo data is devised to be either aggregation of local samples or local samples themselves, I am not sure this method is privacy compliance
>
> Thanks for your suggestion. We would like to clarify that the privacy issue was considered in our initial submission (Figure 5 \(c\)). To clarify the discussion, we have revised our paper (see section 5.3).
> - **Local samples will never be transmitted directly.** In practice, we sample 10 local data and send their mean to the server, as shown in Algorithm 2 of Appendix B.
> - *The initial submission has already considered this issue and proposed `Mixture` (see Sec 4.2) to preserve privacy.*
>     - **The public data collected by the server will not have privacy issues.** All data used in `Mixture` are collected by the server and could be significantly different from local data. Therefore, this operation will not have privacy issues.
>     - **The effectiveness of `Mixture` is justified in Figure 5 \(c\)**: its performance is comparable to and even better than `RSM`, indicating that FedDebias can achieve competitive performance without privacy issues.
>
> > ### 6. From Eq. (2) and (3), I am curious why phi_g is not part of the learnable parameters?
>
> We save the model transmitted from the server as $\phi_g$ and set local model $\phi_i = \phi_g$. Note that $\phi_g$ is to generate global features that help $\phi_i$ to learn unbiased local features (see Algorithm 1). Therefore, we will not train $\phi_g$ here.
>
>
>
> > ### 7. Also, why is x_p \sim D_i in Eq. (3)? Are those typos?
>
> Sorry for the typo. We fixed it using blue lines.
>
> ###### We hope the above responses address your concerns. Please let us know if you have other questions. We’re happy to further answer the questions.

---

> ### Author Response · Authors · 2022-11-11
> **Response to reviewer p1f4 (2/3)**
>
> >  ### 3. the validation (or pseudo) data obviously needs to be balanced -- how do we ensure that if the design of such validation data comes from random or aggregated sample of local data?
>
> Sorry for the misunderstanding.
> - **We set the pseudo-label to 1/C for samples produced by `RSM` without abusing the prior knowledge of the evaluation dataset.** We would like to explain the intuitions below.
>     1. It is a privacy-preserving solution to prevent clients from uploading local datasets labels to infer the exact distribution of the evaluation datasets.
>     2. As we discussed in Figure 6 of Appendix B, such pseudo-data enforces an equivalent classification ability over arbitrary classes, no matter the label distribution of the local datasets.
> - **We added additional experiments to show pseudo-data does not need to be class-balanced in `RSM`.**
>     - Motivated by the comment, we construct 32 pseudo-data in `RSM` in two ways: 1) **Case 1**: only using data from three labels of CIFAR10 to construct pseudo-data. 2) **Case 2**: using data from all labels to construct pseudo-data. The results of two scenarios show label distribution of data to construct pseudo-data will not affect the performance of FedDebias:
>
>
> | Case 1| Case 2 |
> | ---- | ---- |
> | 64.53 $\pm$ 0.39  | 64.47 $\pm$ 0.14 |
>
>
> - *The concern indeed has been tackled by our framework in the initial submission, i.e., the `Mixture` in Sec 4.2.*
>     - **In `Mixture`, the data used to construct pseudo-data is collected by the server and can be of the arbitrary form.** For example, we use CIFAR100 as the public data and CIFAR10 as the local data in our experiments. The pseudo-data created from CIFAR100 could not be class balanced on CIFAR10 because CIFAR100 contains 90 more labels than CIFAR10. Therefore, the concern about balanced class distribution is invalid when using `Mixture`. We show in Figure 5 \(c\) that **the performance of the `Mixture` is comparable and even better to `RSM`**.
>     - **In Equation 4, we designed a new local mix mechanism for `Mixture`, which is essential for `Mixture`** to work well. Our experiments in Figure 5 \(c\) show that without our design (corresponding to K = 0 in Figure 5 \(c\)), the performance of the `Mixture` dropped significantly.
>
>
> > ### 4. Not clear how much validation data is needed for this to work. There should also be experiments plotting the effectiveness of the proposed method vs the amount of validation (pseudo) data.
>
> Sorry for missing details. As suggested by the reviewer, we have detailed the number of pseudo-data in the revised paper. Please note that *the experiments on the varying size of pseudo-data were included in the initial submission (Figure 5 (b)).*
> - **One mini-batch (64 for MNIST and 32 for others) of pseudo-data is enough for FedDebias to achieve competitive performance.** As shown in Figure 5(b), we compare the performance of FedDebias when 1) transmitting one mini-batch of pseudo-data in each round and 2) only transmitting one mini-batch of pseudo-data only at the beginning of the training. Results show that the performance of FedDebias does not drop when reducing the number of pseudo-data.
> - **Transmitting pseudo-data will not significantly increase the communication overhead.** As discussed in Figure 5(b), model performance can be preserved by only transmitting one mini-batch of pseudo-data to clients at the initial training phase, with trivial communication overhead.
> - **Existing methods also need extra communication overhead.** For example, VHL [10] needs to generate virtual data and uses much more virtual data than ours. FedMix [8] needs to transmit pseudo-data generated by `RSM` in each round. Xor mixup [11] needs to upload encoded data in each round. *Our experiments show that FedDebias can achieve better performance with significantly less communication overhead than the above-listed methods (Table 1 & 2, Figure 5 (b), Figure 5 \(c\))*.
>
>
> [11] Shin M J, Hwang C, Kim J, et al. Xor mixup: Privacy-preserving data augmentation for one-shot federated learning[J]. arXiv preprint arXiv:2006.05148, 2020.

---

> ### Author Response · Authors · 2022-11-11
> **Response to reviewer p1f4 (1/3)**
>
> Thanks for your insightful reviews, and we appreciate the valuable suggestions! We’ve revised the manuscript and added additional experiments according to your suggestions. Please kindly find our response to your raised questions below.
>
> > ### 1. There is a complete lack of coverage on a very related suite of personalized FL algorithms specifically designed to deal with this issue
>
> Thank you for pointing that out. We would like to clarify that.
> - **Training a better global model is crucial and worth investigating.** This fact is stated in the seminal works/surveys [1,2,3] that “There are many cases where having a single model is to be preferred, e.g., to provide a model to clients with no data, or to allow manual validation and quality assurance before deployment.” A line of work, like FedProx, SCAFFOLD, MOON, and VHL, was proposed for this purpose. Similar to these works, we also aim to learn an improved global model from local heterogeneous datasets.
> - **Personalized FL is orthogonal to our focus**, as it aims to learn a personalized model that generalizes to the corresponding local data distribution. Note that the quality of the personalized model also depends on the global model, and it may suffer from overfitting and lose its robustness to global data distribution [4,5,6].
>
>
> [1] Yuan, Honglin, et al. "What Do We Mean by Generalization in Federated Learning?." International Conference on Learning Representations. 2021.
>
> [2] Kairouz, Peter, et al. "Advances and open problems in federated learning." Foundations and Trends® in Machine Learning 14.1–2 (2021): 1-210.
>
> [3] Wang J, Charles Z, Xu Z, et al. A field guide to federated optimization[J]. arXiv preprint arXiv:2107.06917, 2021.
>
> [4] Tang, Xueyang, Song Guo, and Jingcai Guo. "Personalized Federated Learning with Clustered Generalization." (2021).
>
> [5] Chen, Hong-You, and Wei-Lun Chao. "On bridging generic and personalized federated learning for image classification." International Conference on Learning Representations. 2021.
>
> [6] Wu S, Li T, Charles Z, et al. Motley: Benchmarking heterogeneity and personalization in federated learning[J]. arXiv preprint arXiv:2206.09262, 2022.
>
> > ### 2. Comparison with ​​Yurochkin, 2019
>
> Thank you for your suggestion. We would like to clarify that.
> - **The proposed method in [7] cannot be used in deep models.** *The algorithm relies on the matching weights and bias in the linear layers; extending to the modern NN architectures (e.g., convolutional layer and normalization layer) is non-trivial.* This observation is also aligned with the code provided by the authors (line 116 of `combine_nets.py`) and in contrast to our method and other baseline methods, e.g., [8.9,10] using modern deep neural networks.
> - **We compared our method with some closely related and recently published SOTA FL algorithms.** For example, our methods surpass strong baselines like FedMix (ICLR 2020) [8], MOON (CVPR 2021) [9], and VHL (ICML 2022) [10]. These SOTA methods have already demonstrated their effectiveness over other baselines; for example, Table 3 of VHL [10] shows that VHL can outperform SCAFFOLD, FedProx, and FedNova.
>
>
> [7] Yurochkin, Mikhail, et al. "Bayesian nonparametric federated learning of neural networks." International Conference on Machine Learning. PMLR, 2019.
>
> [8] Yoon, Tehrim, et al. "FedMix: Approximation of Mixup under Mean Augmented Federated Learning." International Conference on Learning Representations. 2020.
>
> [9] Li, Qinbin, Bingsheng He, and Dawn Song. "Model-contrastive federated learning." Proceedings of the IEEE/CVF Conference on Computer Vision and Pattern Recognition. 2021.
>
> [10] Tang, Zhenheng, et al. "Virtual Homogeneity Learning: Defending against Data Heterogeneity in Federated Learning." arXiv preprint arXiv:2206.02465 (2022).

---

> ### Author Response · Authors · 2022-11-15
> **Look forward to your feedback!**
>
> Dear reviewer p1f4,
>
> Thank you again for your time and effort in reviewing our paper! As the discussion period is ending soon (November 18th), we would like to know if our responses have addressed your concerns. If they haven't, please let us know what is missing so we can address them. Thank you very much.

---

### Official Review · Reviewer_14m7 · 2022-10-24

**Confidence:** 4
**Correctness:** 2
**Technical Novelty And Significance:** 3
**Empirical Novelty And Significance:** 2
**Recommendation:** 5

**Clarity, Quality, Novelty And Reproducibility:**

Clarity:
Some clarification about the problem setting and identified pitfalls of FL can be further added and improved, which is better to understand the empirical results of the proposed method and disentangle the advantages of the introduced components.

Quality:
The technical quality of this work is impressive, but the observations and definitions could be further enhanced by adding some quantitative measures or theoretical analysis.

Novelty:
This work proposed FedDebias, which applies two key de-bias steps in a min-max approach, is technically a new framework considering the existing literature for me.

Reproducibility:
The authors have provided the source code of this paper, which guarantees reproducibility with the detailed experimental setting description.

**Strength And Weaknesses:**

Strength:
1. The focused general problem, i.e., federated learning with heterogeneous data, is significant for federated learning and is very practical.
2. This paper provides further analysis of the pessimistic from learning with heterogeneous data, which are being unable to classify unseen data, local features differ from global features, and local features cannot be accurately distinguished.
3. The proposed FedDebias shows empirical effectiveness in the experiments.

Weaknesses:
1. The motivation of the proposed methods seems to be based on the illustration (e.g., Figure 1), it lacks the empirical justification or some related reference to the direct evidence which can support the existence of the raised issues. Could the author provide more quantitative results to describe the issues, like measuring the local learning bias by some metrics?
2. The three identified "pitfalls of FL" in this paper have been explored or discussed in previous FL literature [1-3], like FedProx, Scaffold, or Moon, which tackle the heterogeneity via analyzing the local client drift or feature level information and proposed corresponding methods. Except for the empirical superiority than previous methods in the experiments (e.g., Table 1), could the author discuss more about the uniqueness of the three identified pitfalls?
3. It is unclear why to consider the domain generalization baseline here. The presentation of the current version is also a little bit confusing as the conventional work in learning with heterogeneity data assumed non-iid data partition for training while keeping the test set unchanged. It is hard to analyze since here the contains two research problems, e.g., federated learning with heterogeneity data and domain generalization, and the proposed methods seem to have superimposed effects on tackling each problem. It could be better if the authors could provide clear and rigorous problem setups in the preliminary (maybe at the beginning of Section 3 or somewhere). This may help to improve the presentation.
4. For the experiments, considering the sensitivity of data partition, could the authors also report the std value in some parts of the experimental results? For the experiment parts, more advanced algorithms in both FL and DG are encouraged to compare, e.g., Scaffold, FedNova, IRM, and VREx, if they are applicable. Since none of the current baseline methods has considered both subproblems.

Other question/comment:
1. It seems to be a typo at the beginning of the 3rd paragraph of Appendix A: "domain generation" -> "domain generalization".
2. The overall presentation of the current version seems to be a little confusing as the tackled problem is both conventional FL with heterogeneous data and FL with domain generalization. It may be better if the author could clarify more about the rigorous problem definition before analysis.
3. It could be better if this work can has some further theoretical analysis about the debiasing effect since section 3 has introduced the definition of local learning bias.
4. Could the authors further discuss the difference between the local bias analyzed in Scaffold with the proposed "local learning bias"?

[1] Li, Tian, et al. "Federated optimization in heterogeneous networks." Proceedings of Machine Learning and Systems 2 (2020): 429-450.
[2] Karimireddy, Sai Praneeth, et al. "Scaffold: Stochastic controlled averaging for federated learning." International Conference on Machine Learning. PMLR, 2020.
[3] Li, Qinbin, Bingsheng He, and Dawn Song. "Model-contrastive federated learning." Proceedings of the IEEE/CVF Conference on Computer Vision and Pattern Recognition. 2021.


**Summary Of The Paper:**

This paper focuses on federated learning with heterogeneous data. Specifically, this paper identifies the negative effects of learning with heterogeneous in the federated framework from a representation learning perspective, which is termed local learning bias in its discussion and analysis. To handle with that challenge, the authors propose FedDebias, to reduce the local learning bias on features and classifiers. They conducted various experiments to verify the effectiveness of the proposed method and claimed FedDebias had outperformed other SOTA FL and domain generalization baselines.

**Summary Of The Review:**

In summary, this paper proposed FedDebias to tackle learning with heterogeneous data. The presentation of the current version is encouraged to improve by making the focused problem setting clearer and adding more discussion except for the empirical effectiveness.

---

> ### Author Response · Authors · 2022-11-11
> **Response to reviewer 14m7 (3/3)**
>
> > ### 6. It could be better if this work can has some further theoretical analysis about the debiasing effect since section 3 has introduced the definition of local learning bias.
>
> Thanks for the suggestion. We would like to clarify that different from the existing min-max methods (e.g., DRO [11] and DANN [12]) and contrastive learning methods [13, 14, 15], our method consider different scenarios and objectives:
> - The objective in DRO [11] aims to maximize the worst-case performance on all domains.
> - DANN [12] is designed to learn invariant features and thus improve the model generalization ability on unseen domains.
> - Contrastive learning methods [13, 14, 15] aim to learn features robust to different augmentation methods to improve the model generalization ability.
> - FedDebias instead is designed to reduce the local learning bias in FL with heterogeneous local data so as to improve the scenario where the global training distribution is identical to the test and where all prior methods/analysis cannot apply. Therefore, it is hard to derive a tight theoretical analysis for FedDebias due to the novel scenarios and algorithm designs.
>     - We believe that the insights derived from our design (the local learning bias and min-max approach) and extensive numerical results are significant to the FL community and may inspire further research of theoretical FL.
>
>
> [11] Levy, Daniel, et al. "Large-scale methods for distributionally robust optimization." Advances in Neural Information Processing Systems 33 (2020): 8847-8860.
>
> [12] Ganin, Yaroslav, et al. "Domain-adversarial training of neural networks." The journal of machine learning research 17.1 (2016): 2096-2030.
>
> [13]Chen, Ting, et al. "A simple framework for contrastive learning of visual representations." International conference on machine learning. PMLR, 2020.
>
> [14] Chuang, Ching-Yao, et al. "Debiased contrastive learning." Advances in neural information processing systems 33 (2020): 8765-8775.
>
> [15] Misra, Ishan, and Laurens van der Maaten. "Self-supervised learning of pretext-invariant representations." Proceedings of the IEEE/CVF Conference on Computer Vision and Pattern Recognition. 2020.
>
> ###### We hope the above responses address your concerns. Please let us know if you have other questions. We’re happy to further answer the questions.

---

> ### Author Response · Authors · 2022-11-11
> **Response to reviewer 14m7 (2/3)**
>
> > ### 3. It is unclear why to consider the domain generalization baseline here. /The presentation of the current version is encouraged to improve by making the focused problem setting clearer
>
> Sorry for the confusion. We would like to clarify (in the text below as well as sec 5.1 with blued lines) that:
> - we aim to tackle the issue of data heterogeneity, a standard but *unsolved* challenge in FL.
> - To justify our solution's effectiveness and superior performance over existing methods, we also consider the recent emerging FL methods adapted from domain generalization (DG) [6,7].
>
>
> [6] Mohri, Mehryar, Gary Sivek, and Ananda Theertha Suresh. "Agnostic federated learning." International Conference on Machine Learning. PMLR, 2019.
>
> [7] Deng, Yuyang, Mohammad Mahdi Kamani, and Mehrdad Mahdavi. "Distributionally robust federated averaging." Advances in Neural Information Processing Systems 33 (2020): 15111-15122.
>
>
>
> > ### 4. For the experiments, considering the sensitivity of data partition, could the authors also report the std value in some parts of the experimental results?
>
> Thanks for your suggestion.
> - Due to the page limitation of the main paper, *we only included results with std for some key experiments in Appendix C.1 in the original submission* (e.g., Table 5), where our method consistently outperforms other methods.
> - The consistent and noticeable gains over other methods can be found in Figure 7 of Appendix B, demonstrating that *our improvement does not come from a random fluctuation*.
>
> > ### 5. For the experiment parts, more advanced algorithms in both FL and DG are encouraged to compare, e.g., Scaffold, FedNova, IRM, and VREx, if they are applicable.
>
> To respond to this concern, we have added some mentioned baselines or explained the infeasible comparison:
> - *In the revised version, we evaluated SCAFFOLD on RotatedMNIST and CIFAR10 datasets in Tables 10 & 11 of Appendix C.2.* Results show that **SCAFFOLD cannot outperform FedAvg on modern deep models**. Similar observations can be found in [1,2,3].
> - **IRM failed to converge in the FL scenarios.** We utilized the IRM implementation from the well-known DomainBed [8]. However,
>     1. extending IRM to FL is non-trivial due to its uncontrolled penalty. In our observation, as the global model may perform poorly on some clients, the penalty of IRM becomes extremely large on some clients after the global aggregation, making it fails to converge.
>     2. the effectiveness of IRM is also questioned in [9], as the authors mentioned: "IRM and its alternatives fundamentally do not improve over standard Empirical Risk Minimization."
> - **VREx cannot be directly used in FL settings**, as we discussed in Appendix A in the original submission. In each optimization step, VREx gathers losses from all domains and uses the std value of losses from all domains to construct the penalty. This will significantly increase the communication overhead and is not applicable in FL for the high communication frequency.
> - **Our framework is orthogonal to FedNova.** Our framework simplifies the comparison and only considers the challenging and well-recognized label-skew data heterogeneity [1,2,10]. We use an equal amount of local update steps for all clients. Therefore our backbone method FedAvg is equivalent to FedNova.
> - **Our method can outperform some closely related and recently published SOTA FL algorithms**. For example, we surpass strong baselines like FedMix (ICLR 2020) [10], MOON (CVPR 2021) [2], and VHL (ICML 2022) [1]. These SOTA methods have already demonstrated their effectiveness over baselines pointed out by the reviewer; for example, Table 3 of [1] shows that VHL can outperform SCAFFOLD, FedProx, and FedNova.
>
> [8] Gulrajani, Ishaan, and David Lopez-Paz. "In Search of Lost Domain Generalization." International Conference on Learning Representations. 2020.
>
> [9] Rosenfeld, Elan, Pradeep Kumar Ravikumar, and Andrej Risteski. "The Risks of Invariant Risk Minimization." International Conference on Learning Representations. 2020.
>
> [10] Yoon, Tehrim, et al. "FedMix: Approximation of Mixup under Mean Augmented Federated Learning." International Conference on Learning Representations. 2020.

---

> ### Author Response · Authors · 2022-11-11
> **Response to reviewer 14m7 (1/3)**
>
> Thanks for your insightful reviews, and we appreciate the valuable suggestions! We’ve revised the manuscript and added additional experiments according to your suggestions. Please kindly find our response to your raised questions below.
>
> > ### 1. The motivation of the proposed methods seems to be based on the illustration (e.g., Figure 1), it lacks the empirical justification or some related reference to the direct evidence which can support the existence of the raised issues.
>
>
> Thank you for pointing that out.
> - To support the motivation, we have provided empirical justification in Figures 2 and 3 of Section 3 in the original submission. To make the motivation clear, in the revised manuscript, **we have added more discussions (blue color) to build a stronger connection between sections 1 and 3.**
> - For further strengthening our motivation, in addition to the existing section 3, **we provide extensive empirical results** in Appendix C.3, pages 19, 20, following the suggestions of Reviewer UKVs. For more details, please refer to Question 1 of [Response to reviewer UKVs](https://openreview.net/forum?id=m_thN8e6qrF&noteId=6qGPc3UvFQT).
>
> > ### 2. The three identified "pitfalls of FL" in this paper have been explored or discussed in previous FL literature [1-3], like FedProx, Scaffold, or Moon, which tackle the heterogeneity via analyzing the local client drift or feature level information and proposed corresponding methods. Except for the empirical superiority than previous methods in the experiments (e.g., Table 1), could the author discuss more about the uniqueness of the three identified pitfalls? / Discussion about local bias of SCAFFOLD and ours.
>
> Thank you for your suggestion. We would like to clarify that **although these prior works also considered data heterogeneity, we have some inherent differences**; we have added more discussions in the revised paper with blue lines to clarify the difference.
> - The different definitions of local bias in FedDebias and other related works.
>     - The definition of local bias in existing methods, like FedProx, SCAFFOLD, and MOON, are insufficient to explain the training dynamics in deep learning.
>         - *FedProx* defines the local drifts as the difference between model parameters $\|\| \omega_g - \omega_i \|\|$.
>         - *SCAFFOLD* uses the gradient difference to model client drifts.
>         - *MOON* moves one step forward and defines local bias using the difference between local and global features on local data.
>     - *FedDebias* instead 1) considers both local feature bias and local classifier bias and 2) in a min-max approach:
>         - The definition of local feature bias in FedDebias comes from two observations in FL training, as we have shown in Figure \(3\): 1) the difference between the global feature and local feature on the same input data; 2) the similarity of the local features on different input data.
>         - FedDebias defines the unbalanced classifier outputs as local classifier bias.
> - **Our method empirically outperforms existing methods by a large margin on deep learning tasks.**
>     - Despite the success of theoretical guarantee (under Lipschitz's smooth condition), recently published papers show FedProx and SCAFFOLD usually have minor improvements on deep models; see Table 1 of our submission, Table 4, 5 of [1] (ICML2022), Table 1 of [2] (CVPR2021), Table 2 of [3] (ICML2021), Table 1 of [4] (ICLR2021), Table 2 of [5] (NeurlPS2021).
>     - Our specially-designed min-max strategy and contrastive loss are essential to improve FL on heterogeneous data. Though MOON is a crucial first step that minimizes the distance between global and local features, its performance gain is still limited due to the improper methodology design (our contribution therein). As shown in Tables 1 and 3, we cannot observe a significant improvement once removing our well-designed min-max process, justifying the significance of our contribution to addressing the local bias.
>
>
>
>
>
> [1] Tang, Zhenheng, et al. "Virtual Homogeneity Learning: Defending against Data Heterogeneity in Federated Learning." arXiv preprint arXiv:2206.02465 (2022).
>
> [2] Li, Qinbin, Bingsheng He, and Dawn Song. "Model-contrastive federated learning." Proceedings of the IEEE/CVF Conference on Computer Vision and Pattern Recognition. 2021.
>
> [3] Yoon, Jaehong, et al. "Federated continual learning with weighted inter-client transfer." International Conference on Machine Learning. PMLR, 2021.
>
> [4] Chen, Hong-You, and Wei-Lun Chao. "On bridging generic and personalized federated learning for image classification." International Conference on Learning Representations. 2021.
>
> [5] Luo, Mi, et al. "No fear of heterogeneity: Classifier calibration for federated learning with non-iid data." Advances in Neural Information Processing Systems 34 (2021): 5972-5984.

---

> ### Author Response · Authors · 2022-11-15
> **Look forward to your feedback!**
>
> Dear reviewer 14m7,
>
> Thank you again for your time and effort in reviewing our paper! As the discussion period is ending soon (November 18th), we would like to know if our responses have addressed your concerns. If they haven't, please let us know what is missing so we can address them. Thank you very much.

---

### Official Review · Reviewer_UkVs · 2022-10-27

**Confidence:** 4
**Correctness:** 2
**Technical Novelty And Significance:** 3
**Empirical Novelty And Significance:** 3
**Recommendation:** 5

**Clarity, Quality, Novelty And Reproducibility:**

- Clarity and quality: Writing is clear and easy to follow, but the statements are not supported well by empirical or theoretical evidence.
- Novelty: The idea of using RSM is not novel but the contrastive loss using the RSM is novel.

**Strength And Weaknesses:**

### Strong points

- Experimental results show non-trivial accuracy gain.
- The idea of enforcing high similarity between the global features of pseudo data and the local features of pseudo data is interesting. Also, using an adversarial projection layer to facilitate effective representation learning by the proposed contrastive loss is also straightforward.

### Weak points

- Toy experiments in Section 3 are not precise enough, so the results in the toy experiments may not support the claims (observations)
    - In Figure 2, the authors plot t-SNE for two independently trained models. Even if we train two deep models with the same data, simply a different initialization can result in different representations. Since the two models are trained with different data distributions, it is not surprising that there are distinct clusters for two independent models. In my opinion, the local model should be fine-tuned from the global model with the local data and observe the representation trends.
    - In a similar vein, the small scope of the local features for both seen and unseen classes is also not surprising since the 5-way classification problem is much easier than the 10-way classification.
- Observation in Figure 3 is a widely known problem in existing works in continual learning.
- For the construction of the pseudo-data, setting gt as $1/C$ may be the abuse of the prior knowledge of evaluation datasets since CIFAR10, CIFAR100, and MNIST have balanced data over classes.


### Questions

- Why the accuracy of FedDebias in Table 1 is different from the accuracy in Table 2??

**Summary Of The Paper:**

- This paper proposes two techniques to alleviate the biased local update due to the data heterogeneity across the clients: 1) generating pseudo data by simply averaging a subset of data and regularizing the local model not to be biased toward skewed classes, 2) employing contrastive loss that facilitates close representation between global model and local model for the same classes, while simultaneously enforcing discriminative representation for different classes.
- Experiments on multiple image classification benchmarks validate the effectiveness of the proposed method over SOTA federated learning and domain generalization algorithms.

**Summary Of The Review:**

At this point, this paper is borderline reject. While This paper is well-written and easy to follow. The objective of this work is clear, but the experiments are hard to understand and not convincing to support the main contribution.

---

> ### Author Response · Authors · 2022-11-11
> **Response to reviewer UKVs (2/2)**
>
> > ### 3. For the construction of the pseudo-data, setting gt as 1/C may be the abuse of the prior knowledge of evaluation datasets since CIFAR10, CIFAR100, and MNIST have balanced data over classes.
>
> Sorry for the misunderstanding.
> - **We set the pseudo-label to 1/C for samples produced by `RSM` without abusing the prior knowledge of the evaluation dataset.** We would like to explain the intuitions below.
>     1. It is a privacy-preserving solution to prevent clients from uploading local datasets labels to infer the exact distribution of the evaluation datasets.
>     2. As we discussed in Figure 6 of Appendix B, such pseudo-data enforces an equivalent classification ability over arbitrary classes, no matter the label distribution of the local datasets.
> - **We added additional experiments to show pseudo-data does not need to be class-balanced in `RSM`.**
>     - Motivated by the comment, we construct 32 pseudo-data in `RSM` in two ways: 1) **Case 1**: only using data from three labels of CIFAR10 to construct pseudo-data. 2) **Case 2**: using data from all labels to construct pseudo-data. The results of two scenarios show label distribution of data to construct pseudo-data will not affect the performance of FedDebias:
>
>
> | Case 1| Case 2 |
> | ---- | ---- |
> | 64.53 $\pm$ 0.39  | 64.47 $\pm$ 0.14 |
>
>
> - *The concern indeed has been tackled by our framework in the initial submission, i.e., the `Mixture` in Sec 4.2.*
>     - **In `Mixture`, the data used to construct pseudo-data is collected by the server and can be of the arbitrary form.** For example, we use CIFAR100 as the public data and CIFAR10 as the local data in our experiments. The pseudo-data created from CIFAR100 could not be class balanced on CIFAR10 because CIFAR100 contains 90 more labels than CIFAR10. Therefore, the concern about balanced class distribution is invalid when using `Mixture`. We show in Figure 5 \(c\) that **the performance of the `Mixture` is comparable and even better to `RSM`**.
>     - **In Equation 4, we designed a new local mix mechanism for `Mixture`, which is essential for `Mixture`** to work well. Our experiments in Figure 5 \(c\) show that without our design (corresponding to K = 0 in Figure 5 \(c\)), the performance of the `Mixture` dropped significantly.
>
> > ### 4. Why does the accuracy of FedDebias in Table 1 differ from the accuracy in Table 2?
>
> Sorry for the confusion caused by our ablation study in Table 2.
>
> Our main evaluation in Table 1 considers `RSM` and transmits pseudo-data in each communication round, following a similar baseline setting in [9]. In the ablation study of Table 2 and Figures 5 (b) \& \(c\), we reduced the number of pseudo-data to 1) justify the communication-efficient design of our FedDebias, and 2) enable a fair comparison with VHL. Results show that the performance of FedDebias did not drop when we reduced the number of pseudo-data.
>
> [9] Yoon, Tehrim, et al. "FedMix: Approximation of Mixup under Mean Augmented Federated Learning." International Conference on Learning Representations. 2020.
>
> ###### We hope the above responses address your concerns. Please let us know if you have other questions. We’re happy to further answer the questions.

---

> ### Author Response · Authors · 2022-11-11
> **Response to reviewer UKVs (1/2)**
>
> Dear reviewer UkVs, thank you for your insightful reviews. We really appreciate the valuable suggestions and comments, and we have added additional experiments accordingly. Please kindly find our response to your raised questions below.
>
> > ### 1. Toy experiments in Section 3 are not precise enough. Since the two models are trained with different data distributions, it is not surprising that there are distinct clusters for two independent models. In my opinion, the local model should be fine-tuned from the global model with the local data and observe the representation trends.
>
> Thanks for your suggestion. Motivated by your comment, we conducted experiments under two challenging scenarios below. We found that the observation of local bias still exists when local models are finetuned from the global model.
> - **Scenario 1**
>     * ***Setting***: For a well-trained global model (high accuracy), we fine-tune it on the local dataset for 10 local epochs.
>     * ***Insights***: The drifts between global and local features are large in this setting. Therefore, our observation in Figure 2 \(b\) holds. However, the local features of different input data are not as similar as we showed in Figure 2 \(c\).
> - **Scenario 2**
>     * ***Setting***: For not well-trained global models (low accuracies, e.g., 29.74%, 38.65%, and 49.28%), we fine-tune them on the local dataset for 10  local epochs.
>     * ***Insights***: Our observations for local feature bias hold (both Figure 2(b) \& \(c\)) and such biases are even more significant. This scenario complements the observation of Scenario 1 and justifies the existence of local feature bias over various learning stages.
>
> We have added the new experimental results and the above discussions in Appendix C.3, pages 19 and 20, in the revised paper.
>
>
> > ### 2. Observation in Figure 3 is a widely known problem in existing works in continual learning.
>
> Thanks for your comment. We agree with the reviewer that the observation in Figures 2 \& 3 (local bias) has been well-studied in CL. However, we would like to clarify that:
> - **CL methods cannot be directly applied to FL problems.** Considering some well-known CL methods:
>     - As a close line of research, the regularization-based methods in CL can be similarly found in FL (e.g., FedProx). However, *these regularization-based methods sometimes are hard to observe a significant performance gain on deep learning tasks*. See Table 1 of our submission, Table 4, 5 of [1] (ICML2022), Table 1 of [2] (CVPR2021), Table 2 of [3] (ICML2021), Table 1 of [4] (ICLR2021), and Table 2 of [5] (NeurlPS2021), for more details.
>     - **Replay-based CL methods cannot be directly applied in our FL scenarios**, as moving data across clients violates privacy, while considering the time-varying local dataset is still an open question [6, 7] and beyond the scope of this work: *combining FL and CL is a long road ahead* [8].
> - **Our design of algorithms is non-trivial and significant to the FL community.**
>     * Our min-max optimization strategy on the decomposed feature-classifier layers is a key step in alleviating the issue of data heterogeneity, as shown in Tables 1 & 2 & 3.
>     * Though some methods like MOON [2] also address the issue by decomposing the feature-classifier layers, the missing ingredient of our min-max strategy leads to a very limited gain (see Table 1).
>
> [1] Tang, Zhenheng, et al. "Virtual Homogeneity Learning: Defending against Data Heterogeneity in Federated Learning." arXiv preprint arXiv:2206.02465 (2022).
>
> [2] Li, Qinbin, Bingsheng He, and Dawn Song. "Model-contrastive federated learning." Proceedings of the IEEE/CVF Conference on Computer Vision and Pattern Recognition. 2021.
>
> [3] Yoon, Jaehong, et al. "Federated continual learning with weighted inter-client transfer." International Conference on Machine Learning. PMLR, 2021.
>
> [4] Chen, Hong-You, and Wei-Lun Chao. "On bridging generic and personalized federated learning for image classification." International Conference on Learning Representations. 2021.
>
> [5] Luo, Mi, et al. "No fear of heterogeneity: Classifier calibration for federated learning with non-iid data." Advances in Neural Information Processing Systems 34 (2021): 5972-5984.
>
> [6] Kairouz, Peter, et al. "Advances and open problems in federated learning." Foundations and Trends® in Machine Learning 14.1–2 (2021): 1-210.
>
> [7] Wang J, Charles Z, Xu Z, et al. A field guide to federated optimization[J]. arXiv preprint arXiv:2107.06917, 2021.
>
> [8] Criado M F, Casado F E, Iglesias R, et al. Non-IID data and Continual Learning processes in Federated Learning: A long road ahead[J]. Information Fusion, 2022, 88: 263-280.

---

> ### Author Response · Authors · 2022-11-15
> **Look forward to your feedback!**
>
> Dear reviewer UkVs,
>
> Thank you again for your time and effort in reviewing our paper! As the discussion period is ending soon (November 18th), we would like to know if our responses have addressed your concerns. If they haven't, please let us know what is missing so we can address them. Thank you very much.

---

### Author Response · Authors · 2022-11-11
**Reply to all reviewers**

We thank all reviewers for their time and efforts in reviewing our paper. We provide the summary of revisions and our response to two questions asked by most reviewers in this thread.

> ### Summary of revisions.

**We revised our paper based on the suggestions of all four reviewers using blue lines.** In detail:
- We added new scenarios (see Question 1 of [Reply to reviewer UkVs](https://openreview.net/forum?id=m_thN8e6qrF&noteId=6qGPc3UvFQT) for more details) to support the existence of the local feature bias in FL, as we discussed in sec 3.
- We added clarification in sec 5.1 to explain why we consider standard FL scenario in this work (i.e., test data has the same distribution as the train data).
- We detailed the experimental setting in the main paper. For example, we included the number of pseudo-data we used in sec 5.1 and sec 5.3.
- We polished the existing discussions about privacy and communication overhead in sec 5.3.
- We added discussions on the local learning bias in our paper and other related works (FedProx, SCAFFOLD, MOON) and show their difference in sec 3.
- We added experimental results of SCAFFOLD on RotatedMNIST and CIFAR10 datasets in Tables 10 & 11 of Appendix C.2. Results demonstrate a marginal improvement of SCAFFOLD compared with FedAvg.
- We added an illustration figure (Figure 15) in Appendix C.4 to demonstrate why our key design of the min-max process alleviates the learning difficulties.
- We fixed some typos pointed out by all four reviewers and polished the manuscript carefully.

> ### Regarding the communication overhead.

We would like to clarify that:

- **Transmitting pseudo-data will not significantly increase the communication overhead.** As we discussed in  Figure 5(b), we can transmit one mini-batch of pseudo-data to clients only at the beginning of training, and the communication overhead is trivial.
- **Existing methods also introduce extra communication overhead** (much more severe than us). For example, VHL [1] needs to generate virtual data, and the paper uses significantly more virtual data than ours. FedMix [2] needs to transmit pseudo-data generated by `RSM` in each round. Xor Mixup [3] needs to upload encoded data in each round. *Our experiments show that FedDebias can achieve better performance with significantly less communication overhead than the above-listed SOTA methods (Table 1, 2, Figure 5 (b), Figure 5 \(c\))*.

[1] Tang, Zhenheng, et al. "Virtual Homogeneity Learning: Defending against Data Heterogeneity in Federated Learning." arXiv preprint arXiv:2206.02465 (2022).

[2] Yoon, Tehrim, et al. "Fedmix: Approximation of mixup under mean augmented federated learning." arXiv preprint arXiv:2107.00233 (2021).

[3] Shin M J, Hwang C, Kim J, et al. Xor mixup: Privacy-preserving data augmentation for one-shot federated learning[J]. arXiv preprint arXiv:2006.05148, 2020.


> ### Regarding privacy issues.

We would like to clarify that:

*The initial submission has already considered this issue and proposed `Mixture` (see Sec 4.2) to preserve privacy.*
- **The public data collected by the server will not have privacy issues.** All data used in `Mixture` are collected by the server and could be significantly different from local data. Therefore, this operation will not have privacy issues.
- **The effectiveness of `Mixture` is justified in Figure 5 \(c\)**: its performance is comparable to and even better than `RSM`, indicating that FedDebias can achieve competitive performance without privacy issues.

---

### Decision · Program_Chairs · 2023-01-20

**Decision:**

Reject

**Justification For Why Not Higher Score:**

NA

**Justification For Why Not Lower Score:**

NA

**Metareview: Summary, Strengths And Weaknesses:**

The paper identifies three major problems that can happen in Federated learning with Heterogeneous data and proposes an novel bias removal strategy which is based on generating pseudo labelled data that debiases the algorithm.

The ideas presented are novel and interesting. However, the problem associated with Federated learning over Heterogeneous data setting is very well know and there are several interesting ideas around.  Merely having better results in a contained heterogeneous data setting on three smallish datasets (only on CV MNIST and CIFAR) is not going to impress the community.  Also, the community does not know that the impressive results is because of a particular way the data was sharded? The hetrogenous setting is also not very well defined. How much hetrogenety is bad vs catastrophic. Clearly, the paper needs a significant evaluation as being an empirical only paper.

The idea of creating pseudo data to debais an algorithm is not surprising but it might have several privacy implications.

**Summary Of Ac-Reviewer Meeting:**

The reviews clearly showed lack of excitement for the paper because merely showing better accuracy in a constrained setting is not an impressive argument that a community can learn from. The ideas presented are interesting but needs significantly more validations. stress testing, and more empirical insights for this paper to be an exciting read for the community.